**EMBO** *reports*

# Cotyledon opening during seedling deetiolation is determined by ABA-mediated splicing regulation

Guiomar Martín [1,2,3✉], Ana Confraria [4], Irene Zapata[3], Alvaro Santiago Larran[3], Julia Irene Qüesta [3] & Paula Duque [1✉]

## Abstract

**During seedling deetiolation, plants adjust their development to expose photosynthetic tissues to sunlight, enabling the transition from heterotrophic to autotrophic growth. While various plant hormones are known to influence this process, the role of abscisic acid (ABA) remains unclear. Here, we reveal that ABA plays a major role in controlling the dynamics of cotyledon aperture during seedling deetiolation. In the dark, ABA accumulates in the cotyledons to effectively repress their opening. However, light exposure reverses this effect, allowing the cotyledons to open. Our findings indicate that ABA-mediated regulation of cotyledon dynamics is accompanied by genome-wide rearrangements in both transcriptional and splicing patterns. We demonstrate that ABA-dependent adjustments of cotyledon and splicing dynamics in response to light depend on the positive role of two splicing factors, RS40 and RS41. Moreover, we identify transcriptional and posttranscriptional mechanisms that control the activity of these proteins. Altogether, this work sheds light on the interplay between light and ABA, highlighting cotyledon opening as a new developmental outcome, and identifying alternative splicing as the underlying layer of gene regulation.**

**Keywords** Alternative Splicing; Abscisic Acid; Photomorphogenesis; Seedling Deetiolation; SR Proteins
**Subject Categories** Plant Biology; RNA Biology; Signal Transduction

## Introduction

Plants that germinate in subterranean darkness grow heterotrophically from seed reserves. Under these conditions, hypocotyls elongate rapidly to reach the soil surface and ensure survival, while the apical hook remains folded and the cotyledons closed to protect the apical meristem. This developmental program is known as skotomorphogenesis and, through a process called seedling deetiolation, it switches to photomorphogenesis once plants reach

the soil surface and perceive sunlight (Arsovski et al, 2012). In turn, photomorphogenic development is characterized by the arrest of hypocotyl elongation, the unfolding of the hook, the opening and expansion of cotyledons, and the synthesis of chlorophyl (Gommers and Monte, 2018). These developmental adjustments are key to transitioning to phototrophic growth.

Decades of extensive research have determined the molecular pathways that direct seedling deetiolation. These studies have mainly focused on the transcriptional control of gene expression (Wu, 2014; Cheng et al, 2021). In the dark, the E3 ubiquitin ligase CONSTITUTIVE PHOTOMORPHOGENIC 1 (COP1), whose activity depends on its interaction with SUPPRESSOR OF PHYA-105 (SPA) proteins, favors the accumulation of PHYTOCHROME INTERACTING FACTORS (PIFs) while preventing the action of ELONGATED HYPOCOTYL 5 (HY5) (Deng et al, 1991; Ponnu and Hoecker, 2021). PIFs and HY5 act, respectively, as negative (Leivar et al, 2008; Leivar and Monte, 2014) and positive (Oyama et al, 1997; Gangappa and Botto, 2016) regulators of light responses. When seedlings reach the soil surface, light perceived by the different families of plant photoreceptors enhances the accumulation of positive regulators of photomorphogenesis, which then reverts the transcriptional program established in darkness (Lee et al, 2007; Leivar et al, 2009).

During the transition from intron-containing precursor messenger RNA (pre-mRNA) to mature mRNA (mRNA), introns are removed and the flanking exons joined. Splice sites define the exon/intron boundaries and are recognized by the spliceosome, a large and dynamic protein complex consisting of five small nuclear ribonucleoprotein particles (snRNPs) and over 200 additional proteins (Chen and Moore, 2015). However, not every potential splice site is recognized every time a gene is transcribed, resulting in the production of more than one mRNA from the same gene via a process known as alternative splicing. The splicing outcome is modulated by splicing regulators such as serine/arginine-rich (SR) proteins (Long and Caceres, 2008) and heterogeneous nuclear ribonucleoproteins (hnRNPs) (Han et al, 2010). These proteins bind to *cis*-regulatory elements located in exonic or intronic sequences of the pre-mRNA, to either promote or inhibit spliceosome recognition of specific splice sites.

SR proteins represent the most extensively studied splicing factors in plants. In *Arabidopsis thaliana*, they are encoded by

[1]GIMM - Gulbenkian Institute for Molecular Medicine, Lisbon, Portugal. [2]Department of Biology, Healthcare and the Environment, Faculty of Pharmacy and Food Sciences, Universitat de Barcelona, Barcelona, Spain. [3]Centre for Research in Agricultural Genomics, Cerdanyola del Vallés, Spain. [4]ITQB NOVA - Instituto de Tecnologia Química e Biológica António Xavier, Universidade Nova de Lisboa, Oeiras, Portugal. ✉E-mail: guiomar.martin@cragenomica.es; paula.duque@gimm.pt

18 genes distributed among six subfamilies, three of which are plant-specific; in addition, there are two genes coding for SR-like proteins (Barta et al, 2010). Most of these splicing factors show dynamic expression (Shikata et al, 2014; Calixto et al, 2018) and phosphorylation (van Bentem et al, 2006; Fluhr, 2008) patterns, being often regulated through alternative splicing (Kalyna et al, 2006). Notably, recent genome-wide studies have revealed that splicing landscapes are subjected to light control during seedling deetiolation (Shikata et al, 2014; Martín, 2023; Hartmann et al, 2016), suggesting an active role for this molecular process in deetiolation. In agreement, a few functional studies have demonstrated that mutations in splicing-related proteins (Shikata et al, 2012; Xin et al, 2017; Yan et al, 2022; Kathare et al, 2022), as well as enhanced expression of particular splice variants (Shikata et al, 2014; Hartmann et al, 2016; Dong et al, 2020; Huang et al, 2022), can alter plant responses to light signals at the seedling stage. Remarkably, among all light-induced physiological responses, these phenotypic studies have thus far only revealed defects in the regulation of hypocotyl length. Despite these significant lines of evidence, we are still far from understanding how alternative splicing transduces light signals during seedling deetiolation.

Seedling deetiolation is known to be regulated by various plant hormones (de Wit et al, 2016; Liu et al, 2017), yet the role of abscisic acid (ABA) in this process remains unclear (Humplík et al, 2017). ABA is pivotal in many aspects of plant development, such as seed dormancy, and in implementing stress responses (Cutler et al, 2010). Early ABA signal transduction involves the hormone's perception by intracellular receptors, triggering derepression of SNF1-related kinases 2 (SnRK2) protein kinases, which then phosphorylate downstream effector proteins (Umezawa et al, 2010). Given its accumulation in dormant seeds and plants facing stressful conditions, functional and molecular studies addressing the biological implications of ABA have focused mainly on these physiological contexts. Consequently, the endogenous regulation of ABA in controlling seedling deetiolation is not fully understood.

The accumulation of ABA under stress is known to regulate the ABSCISIC ACID INSENSITIVE 5 (ABI5) transcription factor to repress seedling post-germinative growth (Lopez-Molina et al, 2001), a developmental stage during which light and ABA signaling pathways interact. First, light availability influences ABA sensitivity, which is enhanced in the dark, where COP1 and PIFs promote ABA-induced post-germinative arrest by positively regulating the activity of ABI5 (Yadukrishnan et al, 2020; Qi et al, 2020). Moreover, the positive regulators of photomorphogenesis HY5, FHY3 and DET1 also influence ABA signaling (Chen et al, 2008; Tang et al, 2013; Xu et al, 2020). These reports, which address the molecular interplay between ABA and light signaling, have used exogenously applied ABA, thus mimicking a physiological scenario in which plants are subjected to abiotic stress.

In this study, we investigated the role of endogenous ABA during seedling deetiolation in non-stressful environments. Our findings reveal that etiolated Arabidopsis seedlings accumulate ABA in the cotyledons, contributing to their closure in the dark. Once the plant perceives a light stimulus, the levels of ABA in this tissue decrease, resulting in the opening of the cotyledons. Importantly, we show that ABA control of cotyledon opening during seedling deetiolation relies on the regulation of the activity of two SR proteins, RS40 and RS41. These results not only uncover a new developmental role for ABA but also extend our knowledge of the molecular interplay between ABA and light signaling by introducing an additional layer of regulation: alternative splicing. Our work demonstrates that transcriptional and posttranslational processes control the function of these proteins to enable cotyledon opening in response to light.

## Results

### Endogenous ABA represses cotyledon opening in etiolated seedlings

Fluorescence analysis of transgenic lines expressing the green fluorescent protein (GFP) reporter under the control of either 35S or the promoter of the RAB18 gene, a classical gene marker for ABA content (Hauser et al, 2017), revealed that the RAB18:GFP signal in etiolated seedlings is restricted mainly to cotyledons and disappears after light exposure (Fig. 1A). This finding was supported by the quantification of RAB18 transcript levels in wild-type (WT) cotyledons during seedling deetiolation (Fig. 1B). Thus, our data reveal that ABA predominantly accumulates in the cotyledons of etiolated seedlings. In agreement, the expression levels of RAB18 and RD29B, another important ABA-responsive gene (Hauser et al, 2017), are reduced in mutant seedlings with a constitutive photomorphogenic phenotype in the dark (Appendix Fig. S1). These results are in line with previous reports indicating that ABA levels decline after light exposure in plants (Weatherwax et al, 1996; Symons and Reid, 2003; Humplík et al, 2015b).

Given these observations, we decided to investigate the role of ABA in etiolated cotyledons. We first examined the cotyledon phenotype of three well-known ABA signaling mutants: snrk2.236 (Fujita et al, 2009), pyl1458 (Gonzalez-Guzman et al, 2012), and abi1-1 (Meyer et al, 1994), all of which are hyposensitive to the action of the hormone. Notably, all three mutants displayed more open cotyledons when compared to WT seedlings (Fig. 1C), indicating that the endogenous ABA in the cotyledons represses their aperture in the dark. In line with this conclusion, light-induced cotyledon opening was repressed by exogenously supplied ABA in a dose-dependent manner (Fig. 1D; Appendix Fig. S2), with this effect being nearly undetectable in the ABA-hyposensitive mutants (Fig. 1E,F).

Together, these results show that during seedling deetiolation endogenous ABA levels are tissue and light-dependent. This pattern unveils a functional role in seedling development. In the dark, ABA accumulates in the cotyledons, preventing them from opening. However, after light exposure, ABA levels decline, allowing for photomorphogenic development of the cotyledons.

### ABA reduces light-mediated transcriptional changes in cotyledons

To investigate how ABA affects light responsiveness at the transcriptional level during seedling deetiolation, we sequenced the mRNA of dark-grown cotyledons and cotyledons exposed to light for three hours (h) in the absence or presence of exogenous ABA (Appendix Fig. S3). The comparison between etiolated cotyledons and those exposed to light in the absence of ABA (dark vs WL) showed that 1,289 genes were downregulated in response to light (47%; WL-down) while 1,426 were upregulated (53%; WL-up)

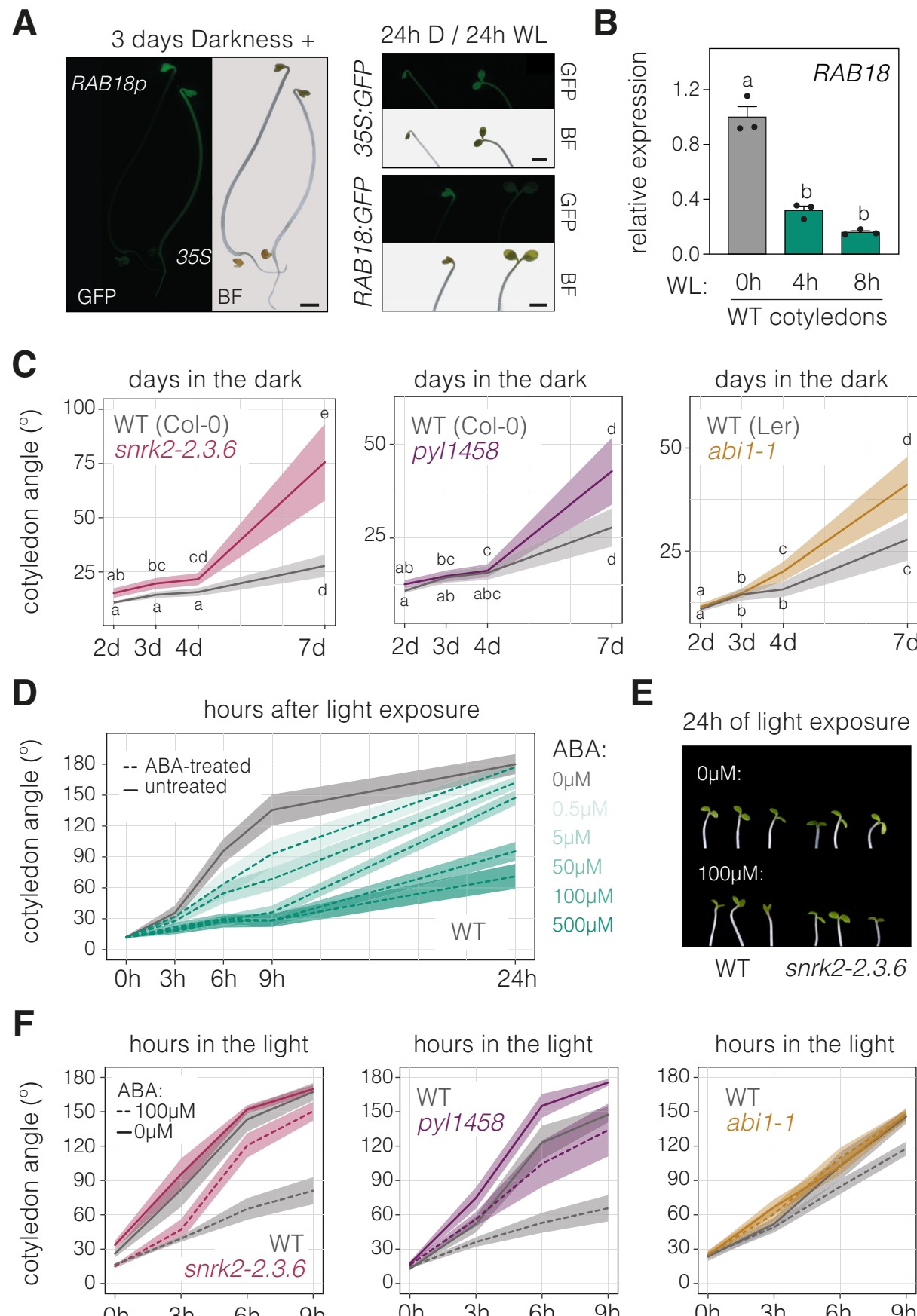

Figure 1. ABA represses cotyledon aperture during seedling deetiolation.

(A) Fluorescence images of seedlings expressing GFP under the control of either the *35S* or the *RAB18* promoter and grown for 3 days in the dark (left) and then exposed for 24 hours (h) to continuous white light (WL) or kept in darkness (D) for the same amount of time (right). Scale bar, 1000 μm. (B) *RAB18* transcript levels analyzed by RT-qPCR in cotyledons of wild-type (WT) seedlings grown in the dark for 3 days (0 h) and then transferred to WL for 4 or 8 h. *PP2A* was used as a reference gene, and expression levels in the dark were set to 1. Data are the means ± SEM of biological triplicates, with different letters indicating statistically significant differences between timepoints by Tukey's multiple comparison test (*P* < 0.05). (C) Quantification of cotyledon opening in the *snrk2-2.3.6*, *pyl1458* and *abi1-1* ABA-hyposensitive mutants and their respective WT seedlings (Col-0 or Ler) grown in the dark for 2, 3, 4, or 7 days (d). At least 70 seedlings per genotype were analyzed. Different letters indicate statistical differences between medians (Dunn's multiple comparison test; *P* < 0.05). (D) Quantification of cotyledon opening in at least 35 WT seedlings grown for 3 days in the dark and then exposed to light in the presence of different concentrations (μM) of ABA. (E) Representative images of 3-day-old Col-0 and *snrk2* triple mutants exposed for 24 h to WL in the absence or presence of 100 μM ABA. (F) Quantification of cotyledon opening in the three ABA-hyposensitive mutants and their respective WTs grown as in (D). At least 40 seedlings per genotype were analyzed. (C, D, F) Thick lines and shaded areas represent, respectively, the median and the 95% confidence interval. A minimum of two biological replicates were performed, all showing similar results. Source data are available online for this figure.

(Dataset EV1). Notably, around two-thirds of these light-regulated genes showed reduced responsiveness to light after 3 h of ABA treatment (Fig. 2A; dark vs WL-ABA). Although exogenous ABA effects should be interpreted with caution, these findings strongly suggest that ABA largely antagonizes the transcriptional changes triggered by light, which is in line with its repressive role in cotyledon opening (see Fig. 1). As exemplified in Appendix Fig. S4, not all of the light-mediated transcriptional changes were subjected to ABA control, indicating that ABA does not produce a universal disruption of light transcriptional regulation.

To gain further insight into ABA control of light signaling, we focused our analysis on genes whose light responsiveness was suppressed by ABA (ABA-reversed; Fig. 2B; see "Methods" for further details). A gene ontology (GO) analysis revealed that genes whose light induction was reversed by ABA are linked to biological processes and cellular compartments associated with organ growth, such as cell division and the cell wall (Fig. 2C; Dataset EV2). Accordingly, the expression of *SAUR* genes, which are established mediators of light-induced cotyledon opening (Dong et al, 2019), aligns with this pattern (Fig. 2D). These results are consistent with our discovery that cotyledon photomorphogenic changes are suppressed by ABA (Fig. 1; Appendix Fig. S2). On the other hand, genes whose light downregulation was reversed by ABA are enriched for ABA-responsive genes, such as the key transcription factors of the ABA signaling pathway *ABI5* and *ABF3* (Fig. 2C; Appendix Fig. S5). The fact that ABA-responsive genes are enriched among our set of cotyledon light-repressed genes reversed by ABA supports our finding that ABA is functionally active in this tissue in the dark.

## ABA represses light-regulated splicing changes in cotyledons

Previous studies have determined how light modifies the splicing landscape of whole etiolated seedlings (Shikata et al, 2014; Hartmann et al, 2016; Martín, 2023). Our splicing analysis of cotyledon samples revealed that light regulation of alternative splicing in etiolated cotyledons coincides with that observed in whole seedlings. First, the 224 genes subjected to alternative splicing in response to light (Dataset EV3) were enriched for splicing-related GO categories (Appendix Fig. S6A; Dataset EV4; Shikata et al, 2014; Hartmann et al, 2016; Martín, 2023). Second, the number of splice variants with retained introns in the dark was higher than in the light (Appendix Fig. S6B; Hartmann et al, 2016; Martín, 2023). Finally, the mRNA splice forms whose abundance

increases in the dark tend to produce unproductive transcripts, and most of them are targeted to the Nonsense-Mediated Decay (NMD) pathway (Appendix Fig. S6C,D; Hartmann et al, 2016; Martín, 2023). These data demonstrate that, as established for whole seedlings, the proportion of unproductive transcripts in dark-grown cotyledons is higher than upon light exposure.

In the light, we identified 35 differential alternative splicing (DAS) events in response to the 3-hour ABA treatment (WL-ABA vs WL; Dataset EV5). Interestingly, ABA-regulated DAS events were also regulated by light exposure (dark vs WL), and dark conditions mimicked the percent of inclusion (PSI) values of ABA-treated light-grown seedlings (Fig. 3A). Moreover, similarly to what we observed for gene expression, the vast majority of the PSI value changes induced by light in cotyledons from dark-grown seedlings (dark vs WL) were reduced in the presence of ABA (Fig. 3B; dark vs WL-ABA). Figure 3C and Appendix Fig. S7 show this ABA reversion of light-mediated splicing changes in two well-known light-regulated DAS events characterized by producing unproductive isoforms in the dark (Petrillo et al, 2014; Hartmann et al, 2018). In agreement with the overlapping splicing patterns found between dark and ABA conditions, ABA-regulated DAS events in the light (WL-ABA vs WL) are enriched for events that disrupt the ORFs when ABA is present (Fig. 3D; unproductive in WL-ABA). Furthermore, the PSI values of these unproductive DAS events are higher in the *upf1upf3* mutant (Fig. 3E), indicating that the mRNAs in which they occur are degraded through the NMD pathway.

These results thus indicate that, compared to light conditions, both ABA and darkness reduce the proportion of productive mRNAs in the cotyledons. Moreover, the observed splicing patterns (Fig. 3A–C) are in line with our phenotypical data showing that ABA blocks light-induced morphogenic changes. This correlation suggested that splicing dynamics are important for controlling cotyledon opening during seedling deetiolation.

## ABA controls cotyledon opening via regulation of the splicing factors RS40 and RS41

To establish a functional link between light regulation of alternative splicing and cotyledon aperture, we screened the SR family of splicing regulators for mutants with defects in the dynamics of cotyledon opening during seedling deetiolation. Strikingly, this screen identified RS40 and RS41, a pair of paralogous SR splicing factors (Kalyna and Barta, 2004), as positive regulators of this process. Cotyledon opening in two independent *rs40 rs41* double

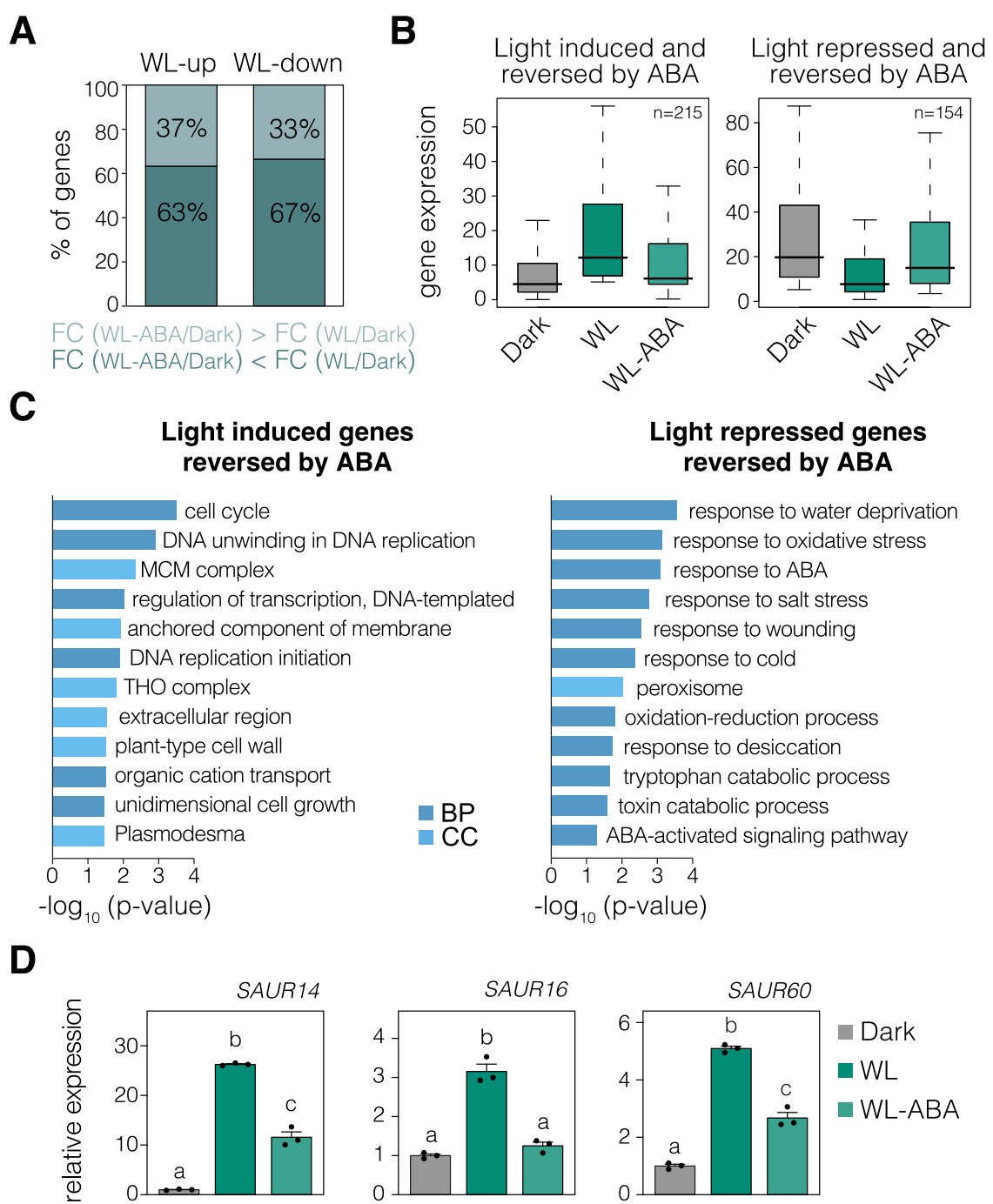

**Figure 2.  ABA reverses light-induced transcriptional changes.**

(A) Percentage of light-repressed or light-induced genes (respectively WL-down and WL-up) whose light responsiveness is lower (FC (WL-ABA/dark) < FC (WL/dark)) or higher (FC (WL-ABA/dark) > FC (WL/dark)) in the presence of ABA. Fold changes (FC) were quantified from our RNA sequencing data for cotyledons extracted from wild-type (WT) seedlings grown for 3 days in the dark and then exposed to continuous white light (WL) for 3 hours in the absence or presence of ABA (100 μM). (B) Distribution of the expression values of genes whose light regulation is reversed by ABA (see "Methods" for details). Boxplots indicate the median (center line), interquartile range (box limits), and minimum and maximum values (whiskers). (C) Enriched gene ontology categories of biological process (BP) and cellular component (CC) for genes defined as light-induced (left) or repressed (right) and reversed by ABA. DAVID P value indicates significance (Fisher's exact test; P < 0.05; Dataset EV2). (D) Transcript levels of genes involved in light-regulated cotyledon opening (Dong et al, 2019) were quantified from our RNA sequencing data. Data are the means ± SEM of biological triplicates and relative to the dark timepoint. Different letters indicate statistically significant differences between conditions by Tukey's multiple comparison test (P < 0.05). Source data are available online for this figure.

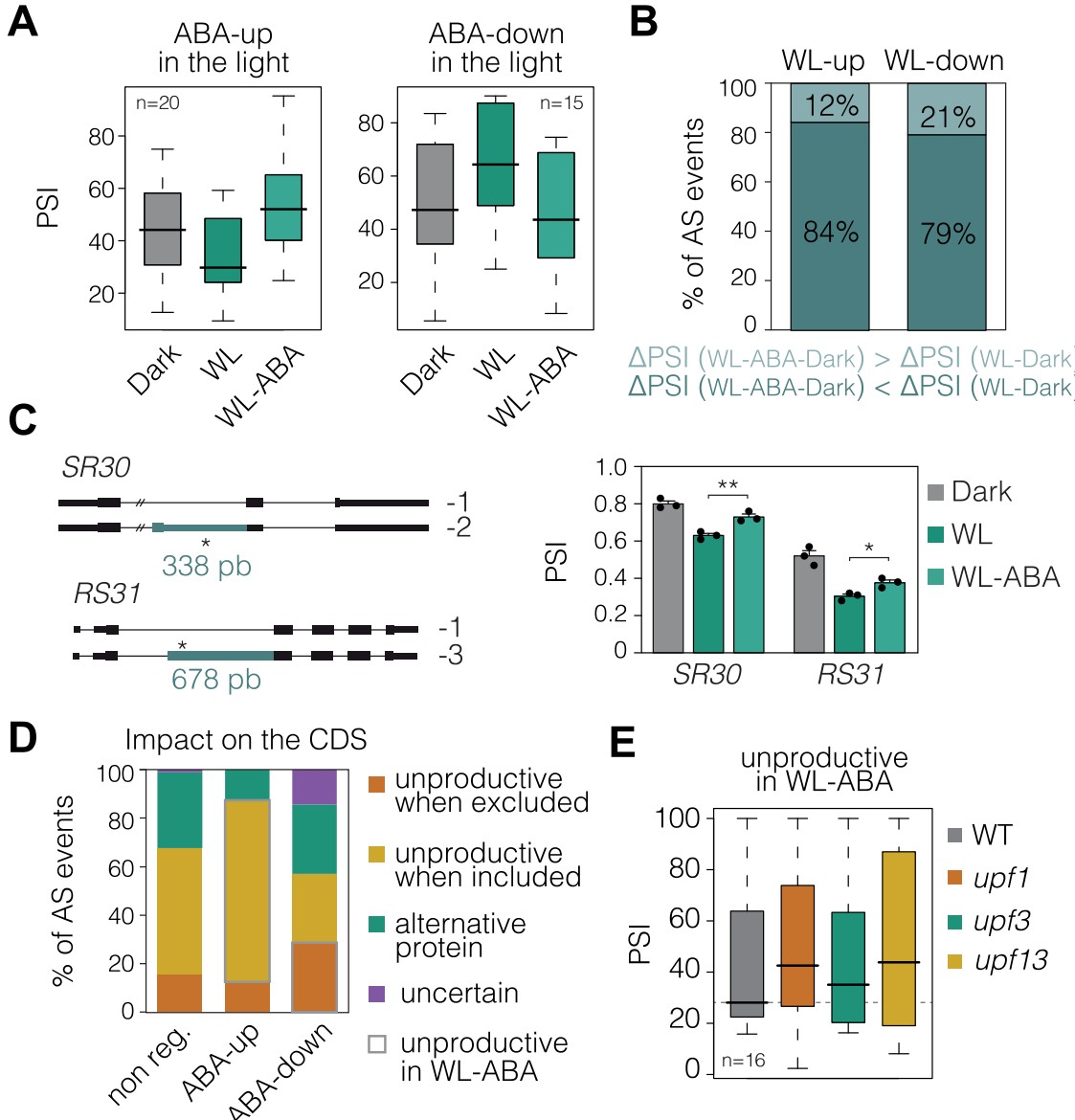

**Figure 3. ABA controls light-regulated splicing changes.**

(A) Percent of inclusion (PSI) values, quantified from our RNA sequencing data, for the differential alternative splicing (DAS) events upregulated (left) and downregulated (right) by ABA in cotyledons from wild-type (WT) seedlings grown for 3 days in the dark and then exposed to continuous white light (WL) for 3 hours (h) in the absence or presence of ABA (100 μM). (B) Percentage of light-induced or light-repressed DAS events (WL-up and WL-down, respectively) whose light responsiveness is lower (WL-ABA < WL) or higher (WL-ABA > WL) in the presence of ABA. (C) RT-PCR analysis of *SR30* (top) and *RS31* (bottom) alternative transcript levels in seedlings grown as in (A), with the exception that seedlings were exposed to WL for 8 h. The bar graphs present PSI values after quantification of the band intensities using the ImageJ software. Data are the means ± SEM of biological triplicates and asterisks indicate statistically significant differences between WL and WL-ABA conditions (Unpaired *t* test: *SR30*, $P = 0.0064$; *RS31*, $P = 0.017$). (D) Percentage of DAS events located in gene coding sequence (CDS) regions that potentially produce unproductive mRNAs or alternative protein isoforms (see "Methods" for details) in three groups of DAS events: non-regulated (non-reg.), and up-, or downregulated by ABA. DAS events that have the ultimate effect of generating unproductive isoforms in response to ABA are indicated as "unproductive in WL-ABA". (E) PSI values of the CDS-located, light-regulated DAS events that generate unproductive transcripts in WT, *upf1*, *upf3*, and *upf1upf3* seedling samples. This quantification was conducted from RNA sequencing data obtained from GSE41432 (Data ref: Drechsel et al, 2013a). (A, E) Boxplots indicate the median (center line), interquartile range (box limits), and minimum and maximum values (whiskers). Source data are available online for this figure.

mutant lines was markedly delayed during seedling deetiolation (Fig. 4A; Appendix Fig. S8A). Phenotyping of two single mutants for each SR protein determined that only the upstream mutation of *RS41* produces a significant delay in cotyledon aperture (Appendix Fig. S9) and that this effect is milder than that of the respective

double mutant (Fig. 4A). We thus concluded that these two proteins act synergistically to positively regulate cotyledon opening during seedling deetiolation. Notably, light-induced alternative splicing changes in the cotyledons were reduced in the *rs40 rs41* double mutant (Appendix Fig. S10).

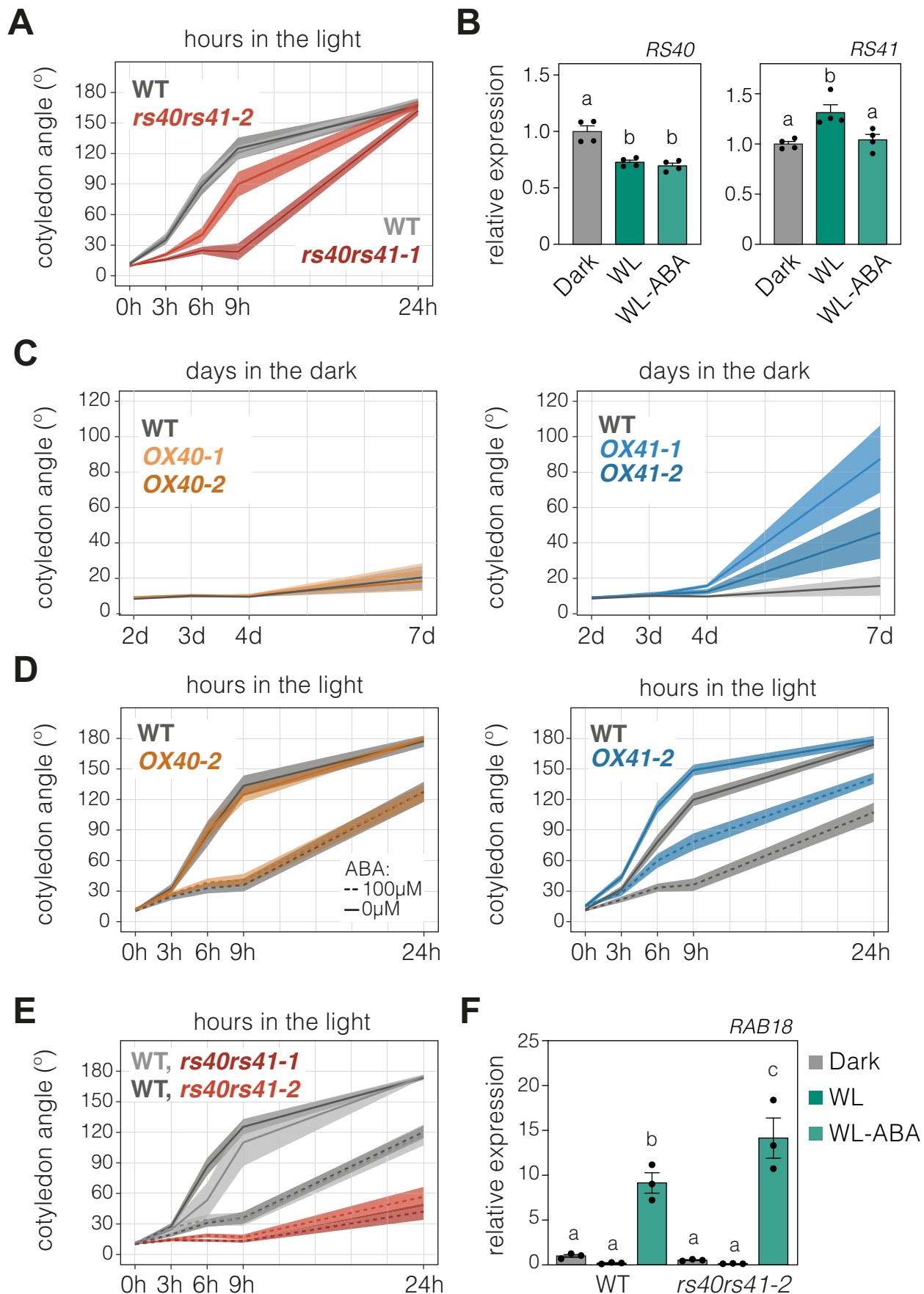

**Figure 4. Splicing factors RS40 and RS41 mediate light and ABA regulation of cotyledon opening.**

(A) Quantification of cotyledon opening in *rs40 rs41* double mutants and their respective wild-type (WT) grown for 3 days in the dark and then exposed to white light (WL) for 3, 6, 9, or 24 hours (h). (B) RT-qPCR quantification of *RS40* and *RS41* transcript levels in cotyledons of WT seedlings grown in the dark for 3 days and then transferred to WL for 3 h in the absence or presence of ABA (100 µM). Data are the means ± SEM of biological quadruplicates. (C) Quantification of cotyledon opening in WT, *RS40*- or *RS41*-overexpressing seedlings grown in the dark for 2, 3 4, or 7 days (d). (D, E) Cotyledon angle values for seedlings of WT, *RS40*- and *RS41*-overexpressing lines (D) or of *rs40 rs41* double mutant lines (E) and their respective WTs grown for 3 days in the dark and then exposed to WL for 3, 6, 9, or 24 h in the absence or presence of ABA (100 µM). (F) RT-qPCR quantification of *RAB18* transcript levels in cotyledons of the WT and *rs40 rs41* mutants grown as in (A) for 8 h. Data are the means ± SEM of biological triplicates. (A, C–E) Thick lines and shaded areas represent, respectively, the median and the 95% confidence interval of at least 65 seedlings. For each phenotypic experiment, a minimum of two biological replicates were performed, all showing similar results. (B, F) *PP2A* was used as a reference gene. For each gene, expression levels in the WT-dark were set to 1. Different letters indicate statistically significant differences between conditions by Tukey's multiple comparison test ($P < 0.05$). Source data are available online for this figure.

Given that RS40 and RS41 are positive regulators of light-induced cotyledon aperture and alternative splicing, both ABA-dependent processes, we hypothesized that the activity of these two proteins is regulated by light in an ABA-dependent manner. We found that in dark-grown cotyledons, *RS40* expression is reduced by light but unaffected by ABA treatment (Fig. 4B). However, the mRNA levels of *RS41* were subtly but consistently enhanced by light only in the absence of ABA (Fig. 4B; Appendix Fig. S11). Therefore, the expression of the latter gene is controlled by both light and ABA.

SR proteins have also long been known to be under extensive alternative splicing control (Palusa et al, 2007; Kalyna and Barta, 2004). In particular, the mRNAs of the Arabidopsis *RS40* and *RS41* genes harbor alternative splicing events that encompass premature stop codons (Kalyna et al, 2006; Iida and Go, 2006). The inclusion levels of these alternative splicing events in our cotyledon samples were also light-regulated; in the dark, the percentage of unproductive mRNAs was higher than in the light but ABA did not interfere with this regulation (Appendix Fig. S12), precluding a functional link between *RS40* and *RS41* splicing patterns and ABA regulation of cotyledon aperture. By contrast, our transcriptional data had suggested that the transcriptional control that light and ABA exert on *RS41*, but not on *RS40*, plays an important role in regulating cotyledon opening during seedling deetiolation. To confirm this, we disrupted the transcriptional regulation of *RS41* and *RS40* by stably expressing each gene under the control of the constitutive *35S* promoter (Appendix Fig. S13). Interestingly, the cotyledons of transgenic lines overexpressing *RS41* but not those of *RS40* overexpressor lines were more open in etiolated seedlings (Fig. 4C; Appendix Fig. S8B). This suggests that *RS41* overexpression overcomes the repressive role that endogenous ABA exerts on cotyledon opening. In accordance, *RS41*-overexpressing seedlings were hyposensitive to exogenous ABA in the light, while seedlings overexpressing *RS40* responded to ABA as WT seedlings (Fig. 4D; Appendix Figs. S8C and S14). In line with the ABA-hyposensitive phenotype of *RS41*-overexpressing plants, *rs40 rs41* double mutants were hypersensitive to ABA at both the phenotypical (Fig. 4E; Appendix Fig. S8A) and molecular (Fig. 4F; Appendix Fig. S15) levels.

Overall, our results indicate that RS40 and RS41 control light-regulated alternative splicing and cotyledon opening during seedling deetiolation in an ABA-dependent manner. Our data also show that the transcriptional control of *RS41* is sufficient to regulate cotyledon aperture downstream of both the light and ABA signals.

## RS40 and RS41 phosphorylation represses cotyledon opening in etiolated seedlings

Extensive data have demonstrated that in eukaryotes, the activity of SR proteins is modulated by phosphorylation (Long and Caceres, 2008). In vivo phosphorylation of SR proteins has also been detected in plants (van Bentem et al, 2006; Wang et al, 2023), although little is known about the biological contexts of this regulation. Here, we examined a potential role for RS40 and RS41 phosphorylation in seedling deetiolation using kinase inhibitors, as previously established (Lin et al, 2022; Haltenhof et al, 2020). To this end, we treated etiolated seedlings with the kinase inhibitor K252a, which is known to mimic light-induced splicing patterns in the dark (Appendix Fig. S16; Hartmann et al, 2016), although the molecular basis for K252a-induced alternative splicing changes remains unknown. Importantly, we found that at least the two K252a-induced alternative splicing changes that we profiled are dependent on RS40 and RS41 (Appendix Fig. S16). Moreover, K252a caused the opening of etiolated cotyledons (Fig. 5A,B; Appendix Fig. S17), indicating that phosphorylation events are pivotal in maintaining cotyledon closure in the dark. Strikingly, the *rs40 rs41* double mutants were insensitive to this effect (Fig. 5A,B), while single mutants were partially affected (Appendix Fig. S18).

These results demonstrate that phosphorylation of RS40 and RS41—or of one of their mRNA targets or upstream regulators—is required in etiolated cotyledons to maintain their distinct closure and splicing profile. Further supporting the significance of posttranslational regulation of RS40 and RS41 at this developmental stage, we observed that K252a enhances cotyledon opening in *RS40*- and *RS41*-overexpressing seedlings (Fig. 5C; Appendix Fig. S19). In fact, *RS40OX* plants behaved like the WT under control conditions, but their cotyledons were more open upon addition of the kinase inhibitor (Fig. 5C; Appendix Fig. S19). Finally, exogenously supplied ABA failed to fully block K252a-induced cotyledon opening (Fig. 5D), consistent with the notion that endogenous ABA in cotyledons acts on RS40 and RS41 through phosphorylation to repress cotyledon aperture in the dark.

## Discussion

A study conducted by Humplík et al, in 2015 revealed a spatio-temporal regulation of ABA content in tomato seedlings during seedling deetiolation (Humplík et al, 2015b). The authors found that in darkness, ABA accumulates predominantly in the

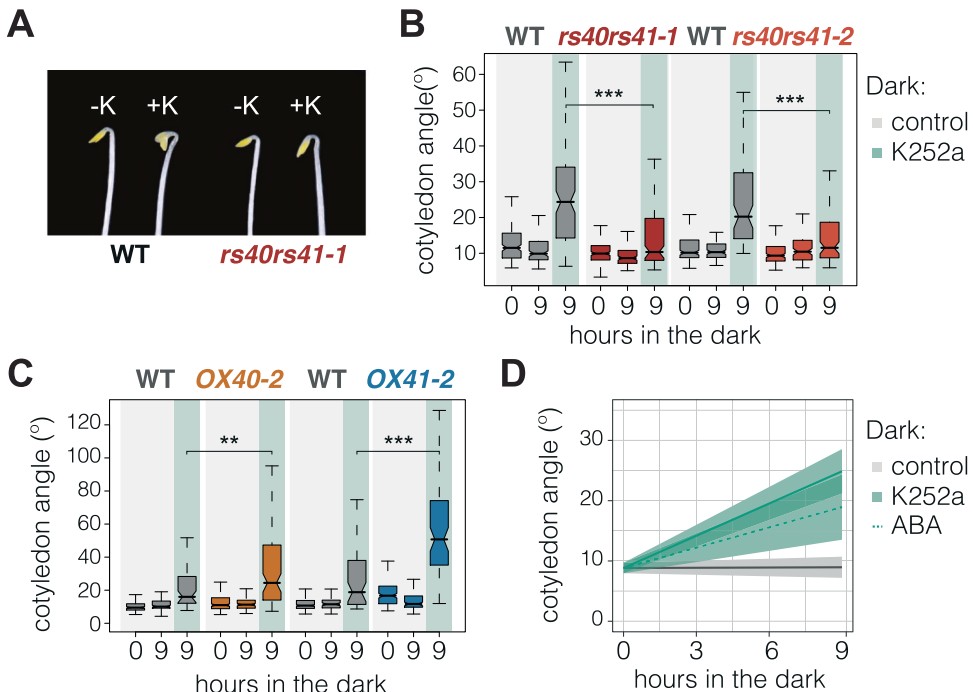

**Figure 5.    Effect of the K252a kinase inhibitor on RS40- and RS41-regulated cotyledon opening.**

(A) Representative images of 3-day-old Col-3 wild-type (WT) and *rs40rs41.1* seedlings exposed for 9 hours (h) to the absence or presence of K252a (1 μM). (B, C) Quantification of cotyledon opening in *rs40 rs41* double mutants (B) and *RS40-* or *RS41*-overexpressing plants (C) as well as in their respective WTs (Col-0 or Col-3; see "Methods" for details) grown as in (A). Boxplots indicate the median of at least 50 seedlings (center line), interquartile range (box limits), and minimum and maximum values (whiskers). Asterisks indicate statistically significant differences between K252a-treated mutants and their respective WTs (Mann–Whitney test: WT vs. *rs40rs41-1* $P < 0.0001$; WT vs. *rs40rs41-2* $P < 0.0001$; WT vs. *OX40-2* $P = 0.0087$; WT vs. *OX41-2* $P < 0.0001$). At least two biological replicates were conducted, all showing similar results. (D) Cotyledon angle values of WT seedlings grown as in (A) and in the presence of ABA (100 μM). Thick lines and shaded areas represent, respectively, the median and the 95% confidence interval of at least 25 seedlings. Source data are available online for this figure.

cotyledons and to a lesser extent in the elongation zone of hypocotyls. This finding aligned with previous research indicating a decrease in ABA levels upon exposure to light in etiolated seedlings of pea and lentil (Weatherwax et al, 1996; Symons and Reid, 2003). Based on these data, as well as on the experimental profiling of hypocotyl length in Arabidopsis and tomato ABA signaling mutants (Barrero et al, 2008; Humplík et al, 2015a), they concluded that in etiolated seedlings endogenous ABA positively affects hypocotyl elongation. This conclusion challenged the traditional view of ABA as a growth inhibitor (Humplík et al, 2017), which is largely based on experiments involving externally supplemented ABA (Lorrai et al, 2018). The authors also argued that the biological function of cotyledon-accumulated ABA might be to inhibit the maturation of chloroplasts and stomatal development, but no data was provided to support this claim. Apart from these initial results, the spatio-temporal regulation of ABA content during seedling deetiolation has not been addressed further. The present study shows that in Arabidopsis ABA levels are elevated in etiolated cotyledons until seedlings are exposed to light (Fig. 1A). This result is supported by the fact that Arabidopsis genes encoding ABA biosynthetic hormones (*AtNCED2*, *3*, *5*, and *9*) are light-repressed (Charron et al, 2009). Interestingly, our data demonstrate that in this tissue endogenous ABA represses cotyledon opening (Fig. 1C,D). This function is in line with the known repressive role of exogenously provided ABA on light-induced cotyledon greening

(Xu et al, 2020; Guan et al, 2014), adding biological insight into the longstanding debate about the relationship between light and ABA (Kraepiel and Miginiac, 1997).

Different studies have implicated SR proteins in mediating ABA-related physiological processes during early seedling development (Laloum et al, 2023, 2018; Albuquerque-Martins et al, 2023; Carvalho et al, 2010; Chen et al, 2013). In particular, the *rs40* and *rs41* loss-of-function mutants are hypersensitive to the repressive effect of externally supplied ABA on seed germination and root elongation (Chen et al, 2013). Apart from the involvement of RS40 and RS41 in the splicing regulation of stress-related genes (Chen et al, 2013) and miRNA biogenesis (Chen et al, 2015), no other functional roles have been identified for these proteins. Here, we show that RS40 and RS41 play a prominent role in promoting light-induced cotyledon opening and alternative splicing during seedling deetiolation (Fig. 6). While distinct splicing profiles of seedlings grown in dark or light conditions, with an accumulation of unproductive splicing isoforms in the dark, had already been reported (Hartmann et al, 2016; Martín, 2023), the involvement of RS40 and RS41 in this regulation remained unknown. Our data demonstrate that both cotyledon opening and light-induced splicing changes in etiolated seedlings are reduced in mutant plants lacking functional RS40 and RS41, indicating that these two proteins play a positive role in mediating light responses. To date, few splicing-related proteins

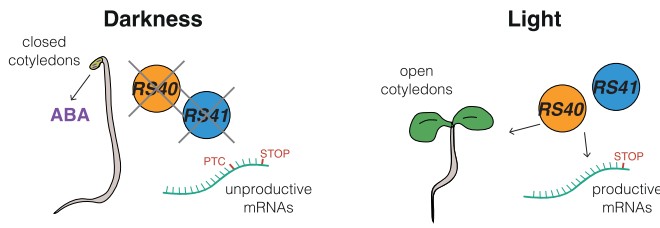

**Figure 6. Proposed model for SR protein regulation of cotyledon opening during seedling deetiolation.**

Cotyledon opening is central to the transition from heterotrophic to autotrophic growth during deetiolation. Our data show that in etiolated seedlings, endogenous ABA prevents the opening of cotyledons and maintains RS40 and RS41 splicing factors inactive. This regulatory mechanism changes the pattern of alternative splicing in response to light, increasing the proportion of functional mRNAs in cotyledons. PTC premature termination codon.

have been implicated in the control of seedling deetiolation. The SWAP1-SFPS-RRC1 ternary complex of RNA-binding proteins was found to coordinate alternative splicing and seedling development during deetiolation downstream of plant photoreceptors (Xin et al, 2017, 2019; Kathare et al, 2022). Notably, our study is the first to link the activity of plant splicing factors to the dynamics of cotyledon opening. Future work will uncover the functional contribution of RS40- and RS41-regulated alternative isoforms in determining cotyledon aperture.

Our results also indicate that the regulation exerted by RS40 and RS41 on cotyledon opening and alternative splicing depends on basal ABA levels. The studies we conducted with loss- and gain-of-function plant lines support a model in which these proteins are inactive in the dark, when ABA content is higher and cotyledons remain closed, and active in the light. Interestingly, experimental data using the K252a kinase inhibitor show that phosphorylation in the dark blocks the capacity of RS40 and RS41 to enhance cotyledon aperture (Fig. 5). Previous studies of the ABA-responsive phosphoproteome identified RS41 among over 100 proteins that were differentially phosphorylated in response to ABA (Wang et al, 2013; Umezawa et al, 2013). Umezawa et al, (2013) found evidence of a phosphopeptide in this protein that is downregulated by ABA, whereas Wang et al, (2013) described this protein as actively phosphorylated by the ABA-activated SnRK2.2, 2.3 and 2.6 kinases. Further work is required to precisely determine the ABA-mediated phosphorylation events that occur in this protein during seedling deetiolation. Importantly, in addition to the posttranslational regulation found to be required for the action of these proteins, our work demonstrates that transcriptional regulation of *RS41* is also crucial for controlling cotyledon opening. Overall, our study underscores a novel role for endogenous ABA in controlling light-mediated cotyledon aperture during seedling deetiolation and links this biological function to the action of the RS40 and RS41 splicing factors. These findings significantly expand our current understanding of ABA and light signals and their molecular interplay, shedding new light on pathways that, so far, have been extensively studied only at the transcriptional level.

# Methods

## Reagents and tools table

| Reagent/ resource | Reference or source | Identifier or catalog number |
| --- | --- | --- |
| **Experimental models** | | |
| *A. thaliana* Col-0 | The European Arabidopsis Stock Center | N60000 |
| *A. thaliana* Col-3 | The European Arabidopsis Stock Center | N8846 |
| *A. thaliana* Ler-0 | The European Arabidopsis Stock Center | CS20 |
| *A. thaliana* rs40-1 | This study | |
| *A. thaliana* rs40-2 | Chen et al, 2015 | |
| *A. thaliana* rs41-1 | Chen et al, 2015 | |
| *A. thaliana* rs41-2 | This study | |
| *A. thaliana* rs40 rs41-1 | This study | |
| *A. thaliana* rs40 rs41-2 | This study | |
| *A. thaliana* 35S::RS40::GFP | This study | |
| *A. thaliana* 35S::RS41::GFP | This study | |
| *A. thaliana* snrk2-2.3.6 | Fujita et al, 2009 | |
| *A. thaliana* pyl1458 | Gonzalez-Guzman et al, 2012 | |
| *A. thaliana* abi1-1 | Meyer et al, 1994 | |
| *A. thaliana* pRAB18::GFP | Hauser et al, 2013 | |
| *Agrobacterium tumefaciens* | | EHA105 |
| **Recombinant DNA** | | |
| Plasmid: pBA002-35S::RS40::GFP | This study | |
| Plasmid: pBA002-35S::RS41::GFP | This study | |
| **Oligonucleotides and other sequence-based reagents** | | |
| Gene expression analysis | | |
| PP2AA3 (AT1G13320) | Shin et al, 2007 | TATCGGATGACGATTCTTCGT |
| | | GCTTGGTCGACTATCGGAATG |
| *RAB18* (AT5G66400) | Díez et al, 2024 | TGGCTTGGGAGGAATGCTTCA |
| | | CCATCGCTTGAGCTTGACCAGA |
| *RD29B* (AT5G52300) | This study | CTTGGCACCACCGTTGGGACTA |
| | | TCAGTTCCCAGAATCTTGAACT |
| *RS40* (AT4G25500) | Cruz et al, 2014 | CCGTTCAAGAAGGAGAGTCC |
| | | TTTCAACTTGGCCATTCTCG |
| *RS41* (AT5G52040) | Cruz et al, 2014 | GGTCAAGGTCGAAGTCAAGC |
| | | GCTCCATCGTATCCTCTTCC |

| Reagent/ resource | Reference or source | Identifier or catalog number |
|---|---|---|
| **Alternative splicing analysis** | | |
| *RS31* (AT3G61860) | Petrillo et al, 2014 | CGGTTGTTCGACAAGTATG |
| | | TGTAGGCTTCAGATTTGAAG |
| *SR30* (AT1G09140) | Petrillo et al, 2014 | GCTATACAGCTCTGTCTCAAG |
| | | TTTCATTTTCAACCAGATATCAC |
| *SRL1* (AT5G37370) | PastDB | TAATCATGTGGAGCCGTGGATG |
| | | GGTGGGAGAAAGTCTCAGC |
| *LPAT1* (AT4G30580) | PastDB | GCTGCAACCCCTGACTCTTC |
| | | GGCAGATTCTCCAAACCCTCG |
| *FRO3* (AT1G23020) | PastDB | GGCGCGTGGTAGACTCGT |
| | | CTGGTTTCCCAAAATAAATCGATTTG |
| *U2AF65A* (AT1G09140) | Petrillo et al, 2014 | GGATGAGCTTAGAGATGATGAGG |
| | | GGCCTGCCACTGGCTGACCATTGG |
| *RS40* (AT4G25500) | PastDB | GAAGCCAGTCTTCTGTGGGAA |
| | | TCCATTCAACACGAAGTCTGCG |
| *RS41* (AT5G52040) | PastDB | AATCATGAAGCCTGTCTTTTGCG |
| | | TTTGTCCACTCAACACGGAGTC |
| **Cloning** | | |
| *RS40* (AT4G25500) | This study | AACTCGAGATGAAGCCAGTCTTCTGTG |
| | | TTGGATCCTAACTCGTCAGCTGGTGG |
| *RS41* (AT5G52040) | This study | AATCATGAAGCCTGTCTTTTGCG |
| | | TTTGTCCACTCAACACGGAGTC |
| **Chemicals, enzymes, and other reagents** | | |
| Murashige & Skoog Medium, including vitamins | Duchefa | M0222 |
| MES | Sigma | M8250 |
| Myo-inositol | Fluka | 57575 |
| ABA | Sigma-Aldrich | A4906 |
| K252a | Cayman | 11338-5 |
| XhoI | NEB | R0146S |
| BamhI | Fermentas | ER0051 |
| InnuPREP Plant RNA kit | Analytik Jena | 845-KS-2060250 |
| DNAse I | Zymo Research | E1010 |
| SuperScript III Reverse Transcriptase | Invitrogen | 18080-044 |
| RNase OUT | Invitrogen | 10777019 |
| Luminaris Color HiGreen qPCR Master Mix, high ROX | Thermo Scientific | K0363 |
| NZYTaq II 2x Green Master Mix | NZYtech | MB35801 |
| Agarose | Lonza | 50004 l |
| **Software** | | |
| GraphPad Prism 8 | https:// www.graphpad.com | |
| National Institutes of Health ImageJ | https://imagej.net/ ij/ | |

| Reagent/ resource | Reference or source | Identifier or catalog number |
|---|---|---|
| Leica LAS X | https://www.leica-microsystems.com/ products/ microscope-software/p/leica-las-x-ls/ | |
| Database for Annotation, Visualization, and Integrated Discovery | Huang et al, 2007 | |
| *Vast-tools* | https://github.com/ vastgroup/vast-tools | |
| **Other** | | |
| PastDB | http://pastdb.crg.eu | |

## Plant materials

The *Arabidopsis thaliana* seeds used in this study include the previously described *snrk2-2.3.6* (Fujita et al, 2009), *pyl1458* (Gonzalez-Guzman et al, 2012) and *abi1-1* (Meyer et al, 1994) mutants, which were kindly provided by E. Baena-González and P. Rodríguez. Seedlings expressing the green fluorescent protein (GFP) under the control of the *RAB18* promoter (Hauser et al, 2013) were obtained from the European Arabidopsis Stock Centre (NASC) in homozygosis (N68523). Single mutants *rs40-1* (Sail_1055_C08; N841686) and *rs41-2* (Salk_066799; N566799) were also obtained from NASC, propagated and genotyped to obtain homozygous mutants. Mutants *rs40-2* (WiscDsLox382G12) and *rs41-1* (Sail_64_C03) were previously reported (Chen et al, 2015) and generously supplied by A. Watcher. As detailed in Appendix Fig. S9, *rs40-1* and *rs41-1* insertional mutants are in the Col-3 background, while *rs40-2* and *rs41-2* are in Col-0. The *rs40rs41-1* and *rs40rs41-2* double mutants were generated by crossing the two mutants of each genetic background and WT siblings from each cross were selected to be used as controls. As specified in the figure legends, each mutant line was always compared to the respective WT background. *RS40OX* and *RS41OX* lines were generated by cloning, respectively, a 1053- and 1074-bp fragment containing the coding sequence region of each gene under the control of the cauliflower mosaic virus (CaMV) 35S strong constitutive promoter in the eGFP-tagged version of the binary pBA002 vector using the XhoI/BamhI restriction sites. The resulting constructs, 35S::RS40-GFP (*OX40*) and 35S::RS41-GFP (*OX41*), were introduced *into Agrobacterium tumefaciens* strain EHA105 and subsequently used for agroinfiltration-mediated transformation of Col-0 seedlings (Clough and Bent, 1998).

## Growth conditions and assessment of cotyledon dynamics

Seeds were surface sterilized and sown on MS medium containing 1x Murashige and Skoog (MS) salts (Duchefa Biochemie), 2.5 mM MES (pH 5.7), 0.5 mM myo-inositol, and 0.8% agar (w/v). After stratification for 4 days at 4 °C in darkness, seeds were submitted to a pulse of 3 h of white light (90 mmol·m$^{-2}$·s$^{-1}$) to induce

germination. For dark time-course experiments, seedlings were then grown in continuous darkness for 2, 3, 4, and 7 days. When assessing cotyledon dynamics during deetiolation, seedlings were grown in the dark for 3 days and then exposed to white light (45 mmol·m$^{-2}$·s$^{-1}$) for 3, 6, 9, and 24 h. Seedlings treated with ABA (Sigma) or the broad-range kinase inhibitor K252a (Cayman) were sown on top of filter paper placed on MS medium and, after 3 days of growth in the dark, transferred to plates supplemented with the respective reagent for different time periods: 3, 4, 6, 8, 9, or 24 h of growth in either dark or white light conditions. The concentration and the time period used in each experiment are detailed in the figure legends. Phenotypic measurements of cotyledon opening were performed using the National Institutes of Health ImageJ software, analyzing at least 30 seedlings per experiment. Phenotypic experiments were repeated at least two times to validate reproducibility. The data are presented as medians and the 95% confidence intervals. In accordance, statistical differences were determined using non-parametric tests (Mann–Whitney U test and Dunn's test) in GraphPad Prism 8. Statistically significant differences were defined as indicated in the figure legends.

## RNA extraction

Total RNA was extracted from dissected *Arabidopsis thaliana* cotyledons using the InnuPREP Plant RNA kit (Analytik Jena BioSolutions) and 1 µg was treated with DNase I (Zymo Research) to remove genomic DNA. cDNA synthesis using the oligo dT primer and the enzyme SuperScript III reverse transcriptase (Invitrogen) was conducted in the presence of RNase Out (Invitrogen). Darkness samples were collected in a dark room under green light. The cDNA was then used to quantify either gene expression or the inclusion of alternatively spliced sequences.

## Gene expression and splicing analysis of individual genes

Gene expression was measured by Reverse Transcription-quantitative Polymerase Chain Reaction (RT-qPCR) using a QuantStudioTM 7 Flex Real-Time PCR System 384-well format and the Luminaris Color HiGreen qPCR Master Mix (Thermo Scientific) on 2.5 µL of cDNA (diluted 1:10) per 10 µL of reaction volume, containing 300 nM of each gene-specific primer. The *PP2A* gene was used for normalization (Shin et al, 2007). Alternative splicing of *SR30*, *RS31*, *SRL1*, *LPAT1*, *FRO3*, *U2AF65A*, *RS40* and *RS41* was quantified from RT-PCRs performed with the NZYTaq II 2x Green Master Mix (NZYtech). Reaction cycles were 95 °C for 3 min (1×), 95 °C for 30 s/58 °C for 30 s/72 °C for 5 min (35X), using primers flanking each alternative sequence. The PCR products were then loaded and run on a 2% agarose gel. To assess gene expression or splicing, a minimum of three biological replicates per condition and/or genotype were analyzed, except for the data shown in Fig. 4B and Appendix Fig. S12, which include quadruplicates. The data are presented as means with standard errors. In accordance, statistical differences were determined using parametric tests (Unpaired *t*-test and Tukey's test) in GraphPad Prism 8. Statistically significant differences were defined as indicated in the figure legends.

## GFP microscopy visualization

Fluorescence images of the *pRAB18-GFP* and *35S-GFP* transgenic plant lines were obtained from 3-day germinated seeds and then transferred to white light (45 mmol·m$^{-2}$·s$^{-1}$) or kept in darkness for 24 h, using the high-sensitivity monochrome camera of a Leica DM6 epifluorescent microscope. Fluorescence images from 3-day-old 35S overexpressing *RS40-GFP* and *RS41-GFP* etiolated seedlings were obtained using a Zeiss 980 Elyra 729 confocal microscope. A z-stack of five slices was captured, and merged images were generated and shown in Appendix Fig. S13. Microscopy experiments were repeated at least two times to confirm reproducibility.

## RNA sample preparation and sequencing

RNA was extracted from 3-day-old WT etiolated seedlings treated or not with ABA (100 µM) for 3 h in complete darkness, with a sample also collected before starting the treatment. These samples were collected in a dark room under green light. Oligo dT, strand-specific libraries from biological triplicates were built and sequenced using HiSeq2500 at the Centre for Genomic Regulation Genomics Unit (Barcelona). An average of 90 million 125-nucleotide paired-end reads were generated per sample.

## Gene expression quantification from RNA sequencing data

Quantification of Arabidopsis transcript expression from our RNA sequencing experiment (GSE273664) and public sequencing data (GSE112662 [Data ref: Pham et al, 2018], GSE164122 [Data ref: Martín and Duque 2021], GSE41432 [Data ref: Drechsel et al, 2013a]) was performed using vast-tools v2.2.2 (Tapial et al, 2017). This tool provides the cRPKM number (corrected-for-mappability reads per kbp of mappable sequence per million mapped reads) for each Arabidopsis transcript. This number is equivalent to the number of mapped reads per million mapped reads divided by the number of uniquely mappable positions of the transcript (Labbé et al, 2012). To identify genes differentially expressed by light, we used the command vast-tools compare_expr with the option -norm to perform a quantile normalization of cRPKM values between dark and light samples. In addition, we filtered out the genes that were not expressed at cRPKM >5 and had read counts >50 across all three replicates of at least one of the two samples compared. Finally, light-regulated genes were defined as those with a fold change of at least 2 between each of the individual replicates from each condition.

## Alternative splicing quantification from RNA sequencing data

We employed vast-tools v2.2.2 (Tapial et al, 2017) to quantify alternative splicing from our sequencing data. This tool maps the RNA sequencing data to the araTha10 library, which is composed of an extended annotation of the Ensemble Plants v31, including all exon–exon and exon–intron junction sequences found in the *Arabidopsis thaliana* genome (Martín et al, 2021). Derived from this mapping, vast-tools quantifies exon skipping (ES), intron retention (IR), as well as alternative donor (ALTD) and acceptor (ALTA) site choices. For all these types of events, vast-tools estimates the percent of inclusion (PSI) of the alternative sequence and associates a quality score to each alternative splicing event, based on the read coverage that sustains its PSI quantification (see https://github.com/vastgroup/vast-tools for details). To define differential alternative splicing (DAS) in response to light or

ABA, we used the vast-tools compare command with the –min_ALT_use 25, -p_IR and -legacy_ALT filters (see https://github.com/vastgroup/vast-tools for details). The first ensures that ALT3 and ALT5 events are located in exons with a minimum PSI of 25 in each compared sample. The second filter eliminates those IR events with a significant imbalance between the two exon–intron junctions ($P < 0.05$; binomial test; Braunschweig et al, 2014). Finally, -legacy_ALT forces up and down sequence inclusion evaluation of all ALT annotated events, not only of the most external spliced sites. Then, light- and ABA-regulated DAS events were defined as those with a $|\Delta\text{PSI}| > 15$ between dark and light samples or ABA-treated and untreated light samples. In addition, to be selected as differentially regulated spliced events, the PSI distribution could not overlap between conditions (-min_range:5).

## Gene ontology enrichment analyses

To identify significantly enriched biological processes, molecular functions and cellular components among the different sets of genes, analyses were performed using the functional annotation classification system DAVID (Huang et al, 2007). For each comparison, only genes with transcripts or DAS events that passed equivalent filters to those used to define differentially expressed or spliced events were used as a background.

## Predicted impact of the alternative sequences on the gene coding sequence

The impact predictions for all differentially spliced events were obtained from the Downloads section of *PastDB* (http://pastdb.crg.eu/wiki/Downloads). Briefly, this tool determines whether an alternative sequence preserves the frame and therefore generates an alternative protein, or whether it contains an in-frame stop codon or frame shift when included or excluded (see Martín et al, 2021 for details). Based on the presence of these disrupting sequences, we classified DAS events as unproductive when included or excluded.

# Data availability

The RNA sequencing data produced in this study were submitted to the Gene Expression Omnibus with the accession number GSE273664.

The source data of this paper are collected in the following database record: biostudies:S-SCDT-10_1038-S44319-025-00495-5.

# Peer review information

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

## Acknowledgements

The authors thank Andreas Wachter, Pedro Rodríguez, and Elena Baena-González for kindly providing the *rs40-2*, *rs41-1*, *pyl1458*, *snrk2-2.3.6* and *abi1-1* mutants, Nil Veciana for technical assistance with the Leica DM6 epifluorescent microscope, and Vera Nunes for excellent plant care. This work was funded by Fundação para a Ciência e a Tecnologia (FCT) through grants PTDC/BIA-BID/30608/2017, PTDC/ASP-PLA/2550/2021 (https://doi.org/10.54499/PTDC/ASP-PLA/2550/2021), CEECIND/02655/2017/CP1428/CT0003 (https://doi.org/10.54499/CEECIND/02655/2017/CP1428/CT0003), 2023.06360.CEECIND/CP2836/CT0009 (10.54499/2023.06360.CEECIND/CP2836/CT0009), UIDB/04551/2020 (https://doi.org/10.54499/UIDB/04551/2020) and UIDP/04551/2020 (https://doi.org/10.54499/UIDP/04551/2020) as well as by the Spanish Ministry of Science and Innovation through grant PID2021-125223NA-I00 (MCIN/AEI/10.13039/501100011033/FEDER). GM was supported by a Ramón y Cajal fellow (RYC2020-030160_I; MCIN/AEI/10.13039/501100011033/FSE) from the Spanish Ministry of Science and Innovation, an EMBO Long-Term Fellowship (ALTF 1576-2016) and a Marie Skłodowska-Curie Individual Postdoctoral Fellowship (EU project 750469). JIQ was a Ramón y Cajal fellow (RYC2021-032539-I). ASL received funding from the AGenT Programme, a European Union's Horizon 2020 research and innovation Marie Skłodowska-Curie (MSCA) COFUND program under grant agreement no. 945043.

## Author contributions

**Guiomar Martín**: Conceptualization; Funding acquisition; Investigation; Writing—original draft; Writing—review and editing. **Ana Confraria**: Investigation; Writing—review and editing. **Irene Zapata**: Investigation; Writing—review and editing. **Alvaro Santiago Larran**: Investigation; Writing—review and editing. **Julia Irene Qüesta**: Funding acquisition; Writing—review and editing. **Paula Duque**: Conceptualization; Funding acquisition; Writing—review and editing.

Source data underlying figure panels in this paper may have individual authorship assigned. Where available, figure panel/source data authorship is

listed in the following database record: biostudies:S-SCDT-10_1038-S44319-025-00495-5.

## Disclosure and competing interests statement
The authors declare no competing interests.

