## [Peer Review File · EMBO Reports]

Cotyledon opening during seedling deetiolation is determined by ABA-mediated splicing regulation

Guiomar Martín, Ana Confraria, Irene Zapata, Alvaro Larran, Julia Qüesta, and Paula Duque

Corresponding author(s): Guiomar Martín (guiomar.martin@cragenomica.es), Paula Duque (paula.duque@gimm.pt)

Review Timeline:

Submission Date:	26th Oct 24
Editorial Decision:	19th Dec 24
Revision Received:	11th Apr 25
Editorial Decision:	6th May 25
Revision Received:	23rd May 25
Accepted:	28th May 25

Editor: Achim Breiling

Transaction Report:

Dear Dr. Duque,

Thank you for the submission of your manuscript to EMBO reports. I have now received the reports from the three referees that were asked to evaluate your study, which can be found at the end of this email.

As you will see, the referees think that these findings are of interest. However, they have several comments, concerns, and suggestions, indicating that a major revision of the manuscript is necessary to allow publication of the study in EMBO reports. As the reports are below, and all the referee concerns need to be addressed, I will not detail them here.

Given the constructive referee comments, I would like to invite you to revise your manuscript with the understanding that the concerns of the referees must be addressed in the revised manuscript and in a detailed point-by-point response. Acceptance of your manuscript will depend on a positive outcome of a second round of review. It is EMBO reports policy to allow a single round of revision only and acceptance of the manuscript will therefore depend on the completeness of your responses included in the next, final version of the manuscript.

- 1) a .docx formatted version of the final manuscript text (including legends for main figures, EV figures and tables), but without the figures included. Figure legends should be compiled at the end of the manuscript text.
- 2) individual production quality figure files as .eps, .tif, .jpg (one file per figure), of main figures and EV figures. Please upload these as separate, individual files upon re-submission.

- 4) a complete author checklist, which you can download from our author guidelines (<https://www.embopress.org/page/journal/14693178/authorguide>). Please insert page numbers in the checklist to indicate where the requested information can be found in the manuscript. The completed author checklist will also be part of the RPF.

- 5) that primary datasets produced in this study (e.g. RNA-seq, ChIP-seq, structural and array data) are deposited in an

appropriate public database. If no primary datasets have been deposited, please also state this in a dedicated section (e.g. 'No primary datasets have been generated and deposited'), see below.

The accession numbers and database should be listed in a formal "Data Availability" section that follows the model below. This is now mandatory (like the COI statement). Please note that the Data Availability Section is restricted to new primary data that are part of this study. This section is mandatory. As indicated above, if no primary datasets have been deposited, please state this in this section

Data availability

8) Regarding data quantification and statistics, please make sure that the number "n" for how many independent experiments were performed, their nature (biological versus technical replicates), the bars and error bars (e.g. SEM, SD) and the test used to calculate p-values is indicated in the respective figure legends (also for EV and Appendix figures). Please also check that all the p-values are explained in the legend, and that these fit to those shown in the figure. Please provide statistical testing where applicable. Please avoid the phrase 'independent experiment', but clearly state if these were biological or technical replicates. Please also indicate (e.g. with n.s.) if testing was performed, but the differences are not significant. In case n=2, please show the data as separate datapoints without error bars and statistics. See also: <http://www.embopress.org/page/journal/14693178/authorguide#statisticalanalysis>

9) Please add scale bars of similar style and thickness to microscopic images, using clearly visible black or white bars (depending on the background). Please place these in the lower right corner of the images themselves. Please do not write on or near the bars in the image but define the size in the respective figure legend.

10) Please also note our reference format:

12) We now use CRedit to specify the contributions of each author in the journal submission system. CRedit replaces the author contribution section. Please use the free text box to provide more detailed descriptions and do NOT provide your final manuscript text file with an author contributions section. See also our guide to authors: <https://www.embopress.org/page/journal/14693178/authorguide#authorshippinguidelines>

13) All Materials and Methods need to be described in the main text using our 'Structured Methods' format, which is required for

all research articles. According to this format, the Methods section should include a Reagents and Tools Table (listing key reagents, experimental models, software, and relevant equipment and including their sources and relevant identifiers), uploaded as separate file, and a Methods section in which we encourage the authors to describe their methods using a step-by-step protocol format with bullet points, to facilitate the adoption of the methodologies across labs. More information on how to adhere to this format as well as downloadable templates (.doc) for the Reagents and Tools Table can be found in our author guidelines (section 'Structured Methods'):

14) Please order the sections like this, using these names:

Title page - Abstract - Keywords - Introduction - Results - Discussion - Methods - Data availability section - Acknowledgements (including funding information) - Disclosure and Competing Interests Statement - References - Figure legends - Expanded View Figure legends

15) Please make sure that all the funding information is also entered into the online submission system and that it is complete and similar to the one in the acknowledgement section of the manuscript text file.

I look forward to seeing a revised form of your manuscript when it is ready.

Yours sincerely,

Referee #1:

This is a very well-conducted and insightful manuscript that investigates the role of ABA in early seedling development. The researchers observed that ABA is present in the cotyledons of dark-grown seedlings but rapidly diminishes after light exposure. ABA levels were indirectly assessed via the output of the RAB18 promoter activity. The finding that ABA is specifically active in cotyledons was further supported by experiments with several ABA-related mutants and the application of exogenous ABA, which revealed that ABA suppresses cotyledon opening in a dose-dependent manner. Additionally, transcriptome analysis showed that ABA can partially counteract light-induced changes in the transcriptome, both at the levels of differential gene expression and alternative splicing. Notably, two splicing factors, RS40 and RS41, exhibited delayed cotyledon opening, and experiments using the global kinase inhibitor K252a led the authors to propose that the phosphorylation of RS40 and RS41 in etiolated seedlings is essential to maintain cotyledon closure.

Major Points for Revision:

Phosphorylation of RS40 and RS41:

The authors conclude that "these results demonstrate the requirement for phosphorylated RS40 and RS41 in etiolated cotyledons to maintain their distinct closure and splicing profile." However, based on the presented experiments, it cannot be excluded that the effects are mediated by the gene product of a differentially expressed or spliced mRNA that becomes phosphorylated, rather than RS40 and RS41 directly. It would greatly enhance the manuscript to provide additional insights into the phosphorylation status of RS40 and RS41 under physiological conditions.

While the use of a broad-range kinase inhibitor (K252a) offers valuable insights, some observed effects may be independent of K252a's action on RS40 and RS41 phosphorylation. The statement, "importantly, we found that at least the two K252a-induced alternative splicing changes that we profiled are dependent on the phosphorylation status of RS40 and RS41," could be changes

to reflect this uncertainty. Since the authors have generated GFP-tagged RS lines, it would be valuable to perform targeted proteomics or Phos-tag gel analysis with these lines under etiolated and de-etiolated conditions, as well as after K252a treatment. This would provide more robust evidence regarding the phosphorylation status of RS40 and RS41 and its physiological significance.

Validation of Alternative Splicing Events:

The authors state that "notably, light-induced alternative splicing changes in the cotyledons were barely perceptible in the rs40 rs41 double mutant." However, upon examining the corresponding supplemental figure, it appears that only two genes were tested by RT-PCR, without quantification, and the experiments were only shown for the rs40 rs41 double mutants, not for the corresponding WT background. To strengthen the claim, alternative splicing validation should include quantification and could also check for additional splicing events. For a more comprehensive analysis, an additional RNA-seq experiment to assess global splicing changes would be ideal. Alternatively, a more extensive RT-PCR panel testing 6-10 alternative splicing events repressed by ABA in de-etiolated seedlings could be provided.

Minor Points:

The labeling of rs double mutants in Figure 4 could be made clearer. For example, using labels such as rs40-1 rs41-1 or rs40;41 #1 instead of rs4041.1 might avoid confusion.

Images of RS mutant and overexpression lines should be presented alongside the quantitative analysis in Figure 4 for better comparison.

Referee #2:

This manuscript presented a study that investigated and provides important insights the complex regulatory interactions between splicing mediated by RS41 and RS40, ABA and light during the seed etiolation process.

Understandably, this is a very complex relationship to illustrate. Many observations have been made. The conclusion made is not supported by the experimental results convincingly. The manuscript could benefit from illustrating the ideas and key findings in an easier and more clear manner for a general audience.

For example:

- 1) The experimental design info is not clearly presented as what samples have been selected, sequenced and analysed. A table to show the experimental design info would be essential at the level of specifying the meta info for individual biological reps.
- 2) Figure 2A, there is some inconsistency between annotations in the main text (D vs WL) and figure annotations (WL-ABA, WL). A clearer explanation of what samples have been used and what comparisons have been made would be very helpful.
- 3) Fig3A PSI for which AS event? 3C labels for bars are missing. 3D is confusing as there is no represented of "unproductive in WL-ABA" in the legend. There is no further discussion what are the specific events of the DAS events are thus it is not clear what the biological interpretation of the result is.
- 4) Figure 4: interquartile range is usually from Q1 to Q3, which should be represented by two dashed lines with solid line (median) in the middle. The figure is very difficult to understand.

Referee #3:

The basic observation of the work reported is that the plant hormone ABA represses opening of cotyledons, whereas exposure to light leads to cotyledon opening and reduction of ABA. The discovery described here is that ABA affects the two splicing factors, RS40 and RS41 which in turn lead to changes in splicing patterns and light stimulated transcription and subsequent opening of cotyledons.

The authors further demonstrate that phosphorylation of the splicing factors is important for the specific activities.

The authors used appropriate mutants to reach the conclusions.

The introduction is very well written and covers all aspects addressed in the experiments.

Methods:

Provide a reference for agroinfiltration.

Which light spectrum was relevant for the cotyledon opening? This should be given.

Results:

p.4 RAB18 is a marker for ABA stimulation/responsiveness not content.

p.5 line 5 omit "functional"

The discovery that ABA levels decline after light exposure is not really novel, so please tone it down in describing the observation

Why was 3hour ABA treatment chosen for the transcriptome analysis, were other time points tested?

p.7 4th line from bottom: why upstream of RS41, what about RS40?

T

he authors should be careful with the interpretation of experiments using exogenous ABA treatment, as this may affect several other processes.

MS ID: EMBOR-2024-60659V1

MS TITLE: *Cotyledon opening during seedling deetiolation is determined by ABA-mediated splicing regulation*

RESPONSE TO REVIEWERS

We would like to thank the three reviewers for their thoughtful and constructive feedback on our manuscript. In response to their comments, we have made several key revisions, including the addition of data to validate alternative splicing events, improved figure labeling for enhanced clarity, and a more detailed explanation of our experimental design and data analysis. While we were unable to provide additional insights into the phosphorylation status of RS40 and RS41 due to technical limitations, we have revised the manuscript to acknowledge the possibility that the observed effects, though dependent on the SR proteins and on phosphorylation, may not necessarily be mediated through direct phosphorylation of these proteins. We are submitting a revised manuscript file, along with a detailed response to each of the reviewers' comments below. We have also collected the source data and made the editorial adjustments requested.

REFEREE #1

This is a very well-conducted and insightful manuscript that investigates the role of ABA in early seedling development. The researchers observed that ABA is present in the cotyledons of dark-grown seedlings but rapidly diminishes after light exposure. ABA levels were indirectly assessed via the output of the RAB18 promoter activity. The finding that ABA is specifically active in cotyledons was further supported by experiments with several ABA-related mutants and the application of exogenous ABA, which revealed that ABA suppresses cotyledon opening in a dose-dependent manner. Additionally, transcriptome analysis showed that ABA can partially counteract light-induced changes in the transcriptome, both at the levels of differential gene expression and alternative splicing. Notably, two splicing factors, RS40 and RS41, exhibited delayed cotyledon opening, and experiments using the global kinase inhibitor K252a led the authors to propose that the phosphorylation of RS40 and RS41 in etiolated seedlings is essential to maintain cotyledon closure.

Major Points for Revision:

COMMENT #1

Phosphorylation of RS40 and RS41:

The authors conclude that "these results demonstrate the requirement for phosphorylated RS40 and RS41 in etiolated cotyledons to maintain their distinct closure and splicing profile." However, based on the presented experiments, it cannot be excluded that the

effects are mediated by the gene product of a differentially expressed or spliced mRNA that becomes phosphorylated, rather than RS40 and RS41 directly. It would greatly enhance the manuscript to provide additional insights into the phosphorylation status of RS40 and RS41 under physiological conditions.

While the use of a broad-range kinase inhibitor (K252a) offers valuable insights, some observed effects may be independent of K252a's action on RS40 and RS41 phosphorylation. The statement, "importantly, we found that at least the two K252a-induced alternative splicing changes that we profiled are dependent on the phosphorylation status of RS40 and RS41," could be changes to reflect this uncertainty. Since the authors have generated GFP-tagged RS lines, it would be valuable to perform targeted proteomics or Phos-tag gel analysis with these lines under etiolated and de-etiolated conditions, as well as after K252a treatment. This would provide more robust evidence regarding the phosphorylation status of RS40 and RS41 and its physiological significance.

RESPONSE

We agree with the Reviewer that, based on the data presented in the paper, we cannot exclude the possibility that a phosphorylation event in a protein other than RS40 and RS41 controls cotyledon aperture and splicing during deetiolation. However, we believe there is consensus that our data from gain- and loss-of-function approaches demonstrate the necessity of RS40 and RS41 for these phosphorylation events to become functionally relevant. Therefore, these phosphorylation events should be considered RS40 and/or RS41-dependent. To address this phosphorylation issue, we performed Phos-tag gel analyses and were able to detect different RS protein bands (see image below). However, the quality of these gels was insufficient for publication, and the results were not consistently reproducible. To improve this approach, we contacted Dr. Andreas Wachter, an expert on the RS subfamily who established the use of K252a in Arabidopsis seedling deetiolation (Hartmann et al. 2016 *The Plant Cell*). He informed us that despite their efforts, his group has also been unsuccessful in detecting band shifts for these proteins using Phos-tag gels. Given these technical limitations, we regret that we were unable to conclusively demonstrate distinct phosphorylation patterns for RS40 and RS41 under our experimental conditions. In response to the Reviewer's insightful comment, we have now incorporated into the manuscript the alternative possibility that the observed effects may result from the phosphorylation of a gene product encoded by a differentially expressed or spliced mRNA, rather than from direct phosphorylation of RS40 or RS41. We also acknowledge the possibility that these effects could arise from phosphorylation of an upstream regulator of RS40 and RS41. In addition to changing the sentence highlighted by the Reviewer—"these results demonstrate the requirement for phosphorylated RS40 and RS41 in etiolated cotyledons to maintain their distinct closure and splicing profile."—we have made several textual modifications throughout the manuscript, including changes to the model depicted in Figure 6. Despite the adjustments, we believe that the data presented provide strong

support for the central conclusions and that the revised manuscript is well-suited for publication in *EMBO Reports*.

COMMENT #2

Validation of Alternative Splicing Events:

The authors state that "notably, light-induced alternative splicing changes in the cotyledons were barely perceptible in the rs40 rs41 double mutant." However, upon examining the corresponding supplemental figure, it appears that only two genes were tested by RT-PCR, without quantification, and the experiments were only shown for the rs40 rs41 double mutants, not for the corresponding WT background. To strengthen the claim, alternative splicing validation should include quantification and could also check for additional splicing events. For a more comprehensive analysis, an additional RNA-seq experiment to assess global splicing changes would be ideal. Alternatively, a more extensive RT-PCR panel testing 6-10 alternative splicing events repressed by ABA in de-etiolated seedlings could be provided.

RESPONSE

We appreciate the Reviewer's comment and acknowledge that our initial conclusion was based on a limited number of genes. We would like to clarify that the supplemental figure originally displayed the splicing pattern of the *rs40 rs41* double mutant alone because the WT pattern was shown in a previous figure (Appendix Figure S7). However, to facilitate direct comparison, we have now included the WT pattern for the two genes profiled throughout the paper and have added quantification of the bands in dark and light samples. Moreover, to

provide a broader perspective on how the *rs40* and *rs41* mutations affect light-regulated alternative splicing, we examined additional light-regulated AS events described in the literature and/or our Dataset EV3. As shown in the updated figure (Appendix Figure S10), our results indicate that light regulation of alternative splicing is not completely abolished in the double mutant but rather reduced for certain genes. We thank the Reviewer's suggestion to test additional genes, as it has enabled us to present more precise data. Accordingly, we have revised the text as follows (lines 238-239): "Notably, light-induced alternative splicing changes in the cotyledons were reduced in the *rs40 rs41* double mutant".

Minor Points:

COMMENT #3

The labeling of rs double mutants in Figure 4 could be made clearer. For example, using labels such as *rs40-1 rs41-1* or *rs40;41 #1* instead of *rs4041.1* might avoid confusion.

RESPONSE

Following the Reviewer's advice, we have renamed the *rs4041* mutants to *rs40rs41* to clarify that it represents a double mutant. This designation aligns with the terminology already used in the main text (*rs40 rs41* double mutants). We have updated Figure 4 and all other relevant figures accordingly. Additionally, we have replaced the dots (.) with dashes (-) when referring to these mutants.

COMMENT #4

Images of RS mutant and overexpression lines should be presented alongside the quantitative analysis in Figure 4 for better comparison.

RESPONSE

We appreciate the Reviewer's suggestion to include images of seedlings as a complement to the quantitative data. In response, we have added representative images of the phenotypic data presented in Figure 4. However, given that this figure already contains several panels (A-F), we determined that the most effective way to incorporate these additional data was to create a new Supplemental figure (Appendix Figure S8). We believe this approach addresses the Reviewer's concern while maintaining the clarity of the main figure.

REFEREE #2

This manuscript presented a study that investigated and provides important insights the complex regulatory interactions between splicing mediated by RS41 and RS40, ABA and light during the seed etiolation process.

Understandably, this is a very complex relationship to illustrate. Many observations have been made. The conclusion made is not supported by the experimental results convincingly. The manuscript could benefit from illustrating the ideas and key findings in an easier and more clear manner for a general audience. For example:

COMMENT #1

1) The experimental design info is not clearly presented as what samples have been selected, sequenced and analysed. A table to show the experimental design info would be essential at the level of specifying the meta info for individual biological reps.

RESPONSE

We agree with the Reviewer that the explanation of the experimental design for the sequenced samples could be clearer. Following the Reviewer's suggestion, we have added a new panel in Appendix Figure S3, which includes a schematic representation and a table detailing the specifications of each sample (Appendix Fig. S3A). This figure is cited on line 160, where the experimental conditions of the RNA sequencing experiment are described.

COMMENT #2

2) Figure 2A, there is some inconsistency between annotations in the main text (D vs WL) and figure annotations (WL-ABA, WL). A clearer explanation of what samples have been used and what comparisons have been made would be very helpful.

RESPONSE

Figure 2A illustrates how ABA treatment influences the expression of genes that are differentially regulated between Dark and WL samples. To achieve these results, we performed sequential comparisons: first, Dark vs WL, and then Dark vs WL-ABA. We recognize that this analysis may not have been sufficiently clear, so we have clarified both the figure and the main text (lines 163-165) to ensure better understanding. The same clarifications have been applied to Figure 3B and line 209, which presents equivalent comparisons for alternative splicing events. Additionally, to improve clarity and maintain

consistency with the figure annotations, we have replaced “D” with “Dark” when referring to Dark samples in the main text.

COMMENT #3

3) Fig3A PSI for which AS event? 3C labels for bars are missing. 3D is confusing as there is no represented of "unproductive in WL-ABA" in the legend. There is no further discussion what are the specific events of the DAS events are thus it is not clear what the biological interpretation of the result is.

RESPONSE

The subset of AS events shown in Figure 3A is indicated at the top of the graph as well as in the figure legend. Since this method of identifying subsets of genes and/or AS events is also used in other panels (Fig. 3B and 3F) and figures (Fig. 2A and 2B), we have maintained this approach, as we believe it is the clearest way to present these data.

We thank the Reviewer for pointing out the missing labels in Figure 3C. These labels have now been added. Additionally, we have clarified the legend of Figure 3 to explain the “unproductive in WL-ABA” events. The same clarification has been applied to Appendix Fig. S6, where unproductive events in darkness are shown.

The molecular validation of the unproductiveness of Dark- and WL-ABA-accumulated transcripts is presented in Appendix Fig. S6D and Fig. 3E, showing that these transcripts tend to be targeted by the nonsense-mediated (NMD) pathway. This is mentioned in the text on lines 196-201 and lines 215-217. Furthermore, on lines 218-219, we link this result to its biological context: *“These results thus indicate that, compared to light conditions, both ABA and darkness reduce the proportion of functional mRNAs in the cotyledons”*. To clarify this connection, we have replaced *“functional”* with *“productive”* in this sentence. Since we consider these unproductive transcripts to be relevant, the AS events individually tested throughout the paper (Fig. 3C, Appendix Fig. S7, S10 and S16) fall into this regulatory category, as detailed in lines 210-211. Moreover, this type of transcripts is depicted in our model (Fig. 6). To provide the appropriate theoretical context for understanding these events, we have revised the manuscript and added a comment in the Discussion section (lines 342-343). Additionally, we have indicated in Dataset EV5 which CDS-located, ABA-regulated AS events in the light generate unproductive transcripts in the presence of ABA.

COMMENT #4

4) Figure 4: interquartile range is usually from Q1 to Q3, which should be represented by two dashed lines with solid line (median) in the middle. The figure is very difficult to understand.

RESPONSE

We thank the Reviewer for this comment. We realized that the information provided in our figure legends was incorrect. Our line plots do not show the interquartile range (IQR) but instead the 95% confidence interval of the median (Median \pm 1.57 X IQR/n). There are two main reasons for this choice. Firstly, this interval corresponds to the notches in boxplots, which provide additional statistical information. Although not a formal test, it is generally considered that if the notches of two boxplots do not overlap, there is “strong evidence” (at the 95% confidence level) that their medians differ (Chambers, John M., William S. Cleveland, Beat Kleiner, and Paul A. Tukey. "Graphical Methods for Data Analysis", 62. Belmont, California: Wadsworth International Group; 1983. ISBN 0-87150-413-8 International ISBN 0-534-98052-X). Secondly, these confidence intervals are narrower than the IQR, making them not only more informative but also improving the visualization of data across multiple genotypes. This method, which represents median values with lines and data distribution with shaded areas, is consistent with approaches used in recent EMBO Journal publications. Similar representations can be found in the following studies: Czarnocka-Cieciura A., et al., *EMBO Journal*, Dec 2024 (Figure 2E or 4G); Soota D., et al., *EMBO Journal*, Nov 2024 (Figure 5B); and Brosey CA., et al., *EMBO Journal*, Feb 2025 (Figure 5G).

REFEREE #3

The basic observation of the work reported is that the plant hormone ABA represses opening of cotyledons, whereas exposure to light leads to cotyledon opening and reduction of ABA. The discovery described here is that ABA affects the two splicing factors, RS40 and RS41 which in turn lead to changes in splicing patterns and light stimulated transcription and subsequent opening of cotyledons.

The authors further demonstrate that phosphorylation of the splicing factors is important for the specific activities.

The authors used appropriate mutants to reach the conclusions.

The introduction is very well written and covers all aspects addressed in the experiments.

Methods:

COMMENT #1

Provide a reference for agroinfiltration.

RESPONSE

We have now included a reference for agroinfiltration at the end of the “Plant Materials” section in the Methods.

COMMENT #2

Which light spectrum was relevant for the cotyledon opening? This should be given.

RESPONSE

We agree with the Reviewer that this is key information to provide. In the “Growth conditions and data representation” section of the Methods, we specify: “After stratification for 4 days at 4°C in darkness, seeds were submitted to a pulse of 3 hours of white light (**90 mmol·m⁻²·s⁻¹**) to induce germination. For dark time course experiments, seedlings were then grown in continuous darkness for 2, 3, 4 and 7 days. When assessing cotyledon dynamics during deetiolation, seedlings were grown in the dark for 3 days and then exposed to white light (**45 mmol·m⁻²·s⁻¹**) for 3, 6, 9 and 24 hours.” However, to improve clarity and make this information easier to locate, we have changed the title of this section and reorganized its content.

Results:

COMMENT #3

p.4 RAB18 is a marker for ABA stimulation/responsiveness not content.

RESPONSE

As suggested by the Reviewer, and to avoid limiting *RAB18* expression to ABA content, we have changed “a classical gene marker for ABA content” to “ABA-marker gene”, as commonly referred to in the literature (Hauser, F., Li, Z., Waadt, R., and Schroeder, J. I. (2017). SnapShot: Abscisic Acid Signaling. Cell 171:1708-1708.e0.).

COMMENT #4

p.5 line 5 omit "functional"

RESPONSE

We thank the Reviewer for pointing this out. We have removed the word “functional”, as requested.

COMMENT #5

The discovery that ABA levels decline after light exposure is not really novel, so please tone it down in describing the observation

RESPONSE

We agree that the observation that ABA levels decline after light exposure is not novel. In fact, we begin the Discussion section by highlighting what is already known about light regulation of ABA levels in plant seedlings. To better emphasize the novelty of our findings, which center on the organ-specificity of this regulation, we have followed the Reviewer’s suggestion and made clarifications in both the Introduction (line 115) and the Results (lines 137-138) sections.

COMMENT #6

Why was 3hour ABA treatment chosen for the transcriptome analysis, were other time points tested?

RESPONSE

The commonly used time points for analyzing molecular responses during seedling deetiolation are 1 hour (early responses) and 3 to 24 hours (late responses) (James M. Tepperman 2004 *Plant Journal*). Due to financial constraints, we were only able to sequence one time point and selected 3 hours to avoid being restricted to early-light responses, which are often transient and not sustained over time (Pablo Leivar 2009 *Plant Cell*). Additionally, the 3-hour time point precedes the drastic phenotypic effects observed after 6 hours of ABA exposure (Figures 1D and 4). To validate this choice, we confirmed by qPCR that classical photomorphogenic marker genes were induced by light at this time point. More importantly, we showed that this time period was sufficient for plants to sense the ABA added to the medium (Appendix Fig. S3). As shown in the paper, additional qPCR and RT-PCR analyses covering other time points further support the genomic observations of this RNA-seq experiment, reinforcing the consistency of the results and conclusions derived from it.

COMMENT #7

p.7 4th line from bottom: why upstream of RS41, what about RS40?

RESPONSE

Among the four single mutants analyzed, two for each RS gene, only the upstream mutation in RS41 (*rs41-1* - Sail 64 C03) results in a drastic delay in the dynamics of cotyledon aperture. To clarify this statement and to indicate that it refers to the results shown in Appendix Fig. S9, we have modified the location of the figure citation (line 235).

COMMENT #8

The authors should be careful with the interpretation of experiments using exogenous ABA treatment, as this may affect several other processes.

RESPONSE

We thank the Reviewer for the thoughtful comment. We agree that exogenous application of ABA can have broad effects on plant physiology, and we carefully considered this in the design of our experiments. To focus on the effects on cotyledon opening and minimize potential confounding factors, we applied transient ABA treatments (3-24 hours), reducing the risk of long-term interference with other ABA-related processes. We also used appropriate controls, including untreated plants and ABA mutants. Notably, we did not observe any other evident ABA-related phenotypes in the RS lines, aside from a slight difference in seed germination rate. However, all of our assays were conducted on plants that had already germinated and grown for 3 days under control conditions before the transient ABA treatment. The application of exogenous ABA is a widely established approach in plant research that has been instrumental in elucidating the hormone's role in a variety of physiological processes and in separating the effects of ABA sensitivity from those of ABA content. Nonetheless, we understand the Reviewer's concern and have now included a note in the manuscript (lines 165-166) highlighting that results from assays with exogenous ABA should be interpreted with caution.

Dear Dr. Martín,

Thank you for the submission of your revised manuscript to our editorial offices. I have now received the reports from two of the three referees that were asked to re-evaluate the study, you will find below. Referee #2 was completely unresponsive to our invitations to re-assess the study. However, going through your p-b-p-response, I consider his/her concerns as adequately addressed.

As you will see, referees #1 and #3 now fully support the publication of your manuscript in EMBO reports. Before we can proceed with formal acceptance, I have these editorial requests I ask you to address in a final revised manuscript:

- Please check again that the number "n" for how many independent experiments were performed, their nature (biological versus technical replicates), the bars and error bars (e.g. SEM, SD) and the test used to calculate p-values is indicated in the respective figure legends. Please also check that all the p-values are explained in the legend, and that these fit to those shown in the figure. Please provide statistical testing where applicable. Please avoid the phrase 'independent experiment' but clearly state if these were biological or technical replicates. Please also indicate (e.g. with n.s.) if testing was performed, but the differences are not significant. In case $n=2$, please show the data as separate datapoints without error bars and statistics. See also:

<http://www.embopress.org/page/journal/14693178/authorguide#statisticalanalysis>

If $n < 5$, please show single datapoints for diagrams. Moreover:

- Please note that the exact p values are not provided in the legends of figures 1B, C; 2D, 3C, 4B, F; 5B, C.
- Please indicate the statistical test used for data analysis in the legends of figures 1B, C; 2D.
- Please note that in figures 1C there is a mismatch between the annotated p values in the figure legend and the annotated p values in the figure file that should be corrected.
- Please note that the box plots need to be defined in terms of minima, maxima, centre, bounds of box and whiskers, and percentile in the legends of figures 2B, 3A, E
- Please note that the box plots need to be defined in terms of minima, maxima, bounds of box and whiskers, and percentile in the legends of figures 5B, C
- Please note that information related to n is missing in the legends of figures 2B, 3A, E.
- Please include the primer information (Dataset EV6) in the 'Reagents and Tools Table' and add callouts accordingly. Then, please remove this dataset from the manuscript files.
- Please remove the instructions and the examples from the final 'Reagents and Tools Table'.
- Please remove the legends of the datasets from the main manuscript text file. Please add these legends on the first TAB of the respective Excel file as a separate TAB.
- Please make sure that all the funding information is also entered into the online submission system and that it is complete and similar to the one in the acknowledgement section of the manuscript text file. Presently, the grant AEI/10.13039/501100011033/FEDER seems missing from the submission system. Please check.
- Please add scale bars of similar style and thickness to all microscopic or photographic images (main and Appendix images), using clearly visible black or white bars (depending on the background). Please place these in the lower right corner of the images themselves. Please do not write on or near the bars in the image but define the size in the respective figure legend.
- During our figure integrity check, we noted a potential blot reuse between panel S10A (upper panels, WT cotyledons Dark and WL) and S16 (upper panels, WT cotyledons Dark and WL). Please check. If the reuse is intentional, please indicate this clearly in the respective figure legends.
- Is there maybe the same WT plant shown in the two panels of S13B? There are several similar features. Please check. Indeed, in the right image there is a splice line visible upon contrast enhancement, indicating that the WT plant might have been pasted in. Please provide the source data (original images) for this panel. If there is a reuse and if this is intentional, please indicate this clearly in the respective figure legend and indicate the splice with a white line.
- In the right upper panel of 1A (35S:GFP, GFP) an edited box with no signal is present, which can also be seen in the source data image (filter enhanced), which shows more squared patches with no signal (see the attached file - panel1A-GFP). Could you please comment?

In addition, I would need from you uploaded separately:

- a short, two-sentence summary of the manuscript (not more than 35 words).
- two to four short (!) bullet points highlighting the key findings of your study (two lines each).

- a schematic summary figure as separate file that provides a sketch of the major findings (not a data image) in jpeg or tiff format (with the exact width of 550 pixels and a height of not more than 400 pixels) that can be used as a visual synopsis on our website.

Best,

Referee #1:

The authors have addressed the concerns raised by this reviewer in previous submissions. Although they unfortunately were unable to determine the phosphorylation status of RS40 and RS41, they presented their arguments in a very convincing manner. As suggested, the text has been revised to moderate the tone of the conclusions. The second main criticism regarding the limited number of AS events analyzed has also been addressed by quantifying four additional AS events in WT and mutants under dark and light conditions. Therefore, all concerns have been satisfactorily addressed.

Referee #3:

The authors have taken into account most comments of the reviewers and they have revised the manuscript appropriately.

This is a very well written manuscript reporting novel results in the field of plant science with a general interest in biological regulatory mechanisms.

Editorial requests:

- Please check again that the number "n" for how many independent experiments were performed, their nature (biological versus technical replicates), the bars and error bars (e.g. SEM, SD) and the test used to calculate p-values is indicated in the respective figure legends. Please also check that all the p-values are explained in the legend, and that these fit to those shown in the figure. Please provide statistical testing where applicable. Please avoid the phrase 'independent experiment' but clearly state if these were biological or technical replicates. Please also indicate (e.g. with n.s.) if testing was performed, but the differences are not significant. In case $n=2$, please show the data as separate datapoints without error bars and statistics.

See also:

<http://www.embopress.org/page/journal/14693178/authorguide#statisticalanalysis>

We have revised all figure legends and made the appropriate changes.

If $n < 5$, please show single datapoints for diagrams. Moreover:

- Please note that the exact p values are not provided in the legends of figures 1B, C; 2D, 3C, 4B, F; 5B, C.

Multiple comparison tests are performed in Figures 1B, 1C, 2D, 4B, 4F, which involves comparisons of each sample with every other sample, resulting in a complex matrix of p-values. In these figure legends, we indicate the statistical criterion used to define significant differences (denoted by different letters): "different letters indicate statistical differences between medians (Dunn's multiple comparison test; $P < 0.05$)." Figure 3C, 5B and 5C show the results of paired comparisons, and we now indicate the exact p-values in the corresponding figure legends.

- Please indicate the statistical test used for data analysis in the legends of figures 1B, C; 2D.

We believe this information is already provided:

Figure 1B: Tukey's multiple comparison test

Figure 1C: Dunn's multiple comparison test

Figure 2D: Tukey's multiple comparison test

- Please note that in figures 1C there is a mismatch between the annotated p values in the figure legend and the annotated p values in the figure file that should be corrected.

Corrected.

- Please note that the box plots need to be defined in terms of minima, maxima, centre, bounds of box and whiskers, and percentile in the legends of figures 2B, 3A, E

This information is now provided.

- Please note that the box plots need to be defined in terms of minima, maxima, bounds of box and whiskers, and percentile in the legends of figures 5B, C

This information is now provided.

- Please note that information related to n is missing in the legends of figures 2B, 3A, E. This information is now provided.

- Please include the primer information (Dataset EV6) in the 'Reagents and Tools Table' and add callouts accordingly. Then, please remove this dataset from the manuscript files.

Done

- Please remove the instructions and the examples from the final 'Reagents and Tools Table'.

Done

- Please remove the legends of the datasets from the main manuscript text file. Please add these legends on the first TAB of the respective Excel file as a separate TAB.

Done

- Please make sure that all the funding information is also entered into the online submission system and that it is complete and similar to the one in the acknowledgement section of the manuscript text file. Presently, the grant AEI/10.13039/501100011033/ FEDER seems missing from the submission system. Please check.

We have confirmed that this grant has been entered into the system with its respective ID: PID2021-125223NA-I00.

- Please add scale bars of similar style and thickness to all microscopic or photographic images (main and Appendix images), using clearly visible black or white bars (depending on the background). Please place these in the lower right corner of the images themselves. Please do not write on or near the bars in the image but define the size in the respective figure legend.

We have ensured that the scale format shown in main and Appendix images is equal.

- During our figure integrity check, we noted a potential blot reuse between panel S10A (upper panels, WT cotyledons Dark and WL) and S16 (upper panels, WT cotyledons Dark and WL). Please check. If the reuse is intentional, please indicate this clearly in the respective figure legends.

Yes, this was intentional, as the Dark and WL conditions in Fig S16 represent control conditions. We have now clarified this in the legend of Appendix Figure 16.

- Is there maybe the same WT plant shown in the two panels of S13B? There are several similar features. Please check. Indeed, in the right image there is a splice line visible upon contrast enhancement, indicating that the WT plant might have been pasted in. Please provide the source data (original images) for this panel. If there is a reuse and if this is intentional, please indicate this clearly in the respective figure legend and indicate the splice with a white line.

Yes, the WTs shown in the two panels of S13B are the same, as transgenic plants in this figure were grown simultaneously in the greenhouse using the same set of WT plants as controls. We have now clarified this in the figure legend.

- In the right upper panel of 1A (35S:GFP, GFP) an edited box with no signal is present, which can also be seen in the source data image (filter enhanced), which shows more squared patches with no signal (see the attached file - panel1A-GFP). Could you please comment?

Yes, the acquisition process for this image was done in sections using the Leica DM6 epifluorescent microscope. To optimize both acquisition time and file size, regions of the slide that did not contain seedlings were not selected for imaging. The empty squares visible in the panel indicate sections of the sample that were intentionally excluded from image capture because they contained no relevant material.

In addition, I would need from you uploaded separately:

- a short, two-sentence summary of the manuscript (not more than 35 words):

ABA prevents cotyledon opening in darkness by modulating splicing through SR proteins RS40 and RS41. Light reduces ABA levels, releasing this repression to allow photomorphogenic cotyledon development.

- two to four short (!) bullet points highlighting the key findings of your study (two lines each)

- Endogenous ABA represses cotyledon opening in dark-grown seedlings.

- ABA inhibits light-induced transcriptional and splicing changes in cotyledons.

- SR proteins RS40 and RS41 mediate ABA's effects on splicing and cotyledon dynamics.

- Phosphorylation is required for ABA-mediated splicing regulation of cotyledon opening.

- a schematic summary figure as separate file that provides a sketch of the major findings (not a data image) in jpeg or tiff format (with the exact width of 550 pixels and a height of not more than 400 pixels) that can be used as a visual synopsis on our website.

Uploaded

Guiomar Martín
Centre for Research in Agricultural Genomics (CRAG)
Plant Molecular Biology
Rua da Quinta Grande, 6
Oeiras 2780-156
Spain

Dear Dr. Martín,

I am very pleased to accept your manuscript for publication in the next available issue of EMBO reports. Thank you for your contribution to our journal.

Yours sincerely,
